# Engineering Challenges of Solution and Slurry-Phase Chemical Hydrogen Storage Materials for Automotive Fuel Cell Applications

**DOI:** 10.3390/molecules26061722

**Published:** 2021-03-19

**Authors:** Troy Semelsberger, Jason Graetz, Andrew Sutton, Ewa C. E. Rönnebro

**Affiliations:** 1Los Alamos National Laboratory, Los Alamos, NM 87545, USA; 2HRL Laboratories, LLC., Malibu, CA 90265, USA; jagraetz@hrl.com; 3Oak Ridge National Laboratory, Oak Ridge, TN 37831, USA; suttonad@ornl.gov; 4Pacific Northwest National Laboratory, Richland, WA 99352, USA; Ewa.Ronnebro@pnnl.gov

**Keywords:** hydrogen storage, ammonia borane, alane, fuel cells, engineering, borazine, diborane

## Abstract

We present the research findings of the DOE-funded Hydrogen Storage Engineering Center of Excellence (HSECoE) related to liquid-phase and slurry-phase chemical hydrogen storage media and their potential as future hydrogen storage media for automotive applications. Chemical hydrogen storage media other than neat liquid compositions will prove difficult to meet the DOE system level targets. Solid- and slurry-phase chemical hydrogen storage media requiring off-board regeneration are impractical and highly unlikely to be implemented for automotive applications because of the formidable task of developing solid- or slurry-phase transport systems that are commercially reliable and economical throughout the entire life cycle of the fuel. Additionally, the regeneration cost and efficiency of chemical hydrogen storage media is currently the single most prohibitive barrier to implementing chemical hydrogen storage media. Ideally, neat liquid-phase chemical hydrogen storage media with net-usable gravimetric hydrogen capacities of greater than 7.8 wt% are projected to meet the 2017 DOE system level gravimetric and volumetric targets. The research presented herein is a collection of research findings that do not in and of themselves warrant a dedicated manuscript. However, the collection of results do, in fact, highlight the engineering challenges and short-comings in scaling up and demonstrating fluid-phase ammonia borane and alane compositions that all future materials researchers working in hydrogen storage should be aware of.

## 1. Introduction

An integrated fuel cell power system for automotive application is a topic that has continued interest because of its potential for increasing fuel efficiency. With ever growing concerns regarding environmental pollution, energy security, and future oil supplies, the global community is seeking non-petroleum-based, regenerable hydrogen storage media. The most promising hydrogen storage media will be the fuel that has the greatest impact on society. The major impact areas include well-to-wheel greenhouse gas emissions, non-petroleum feed stocks, well-to-wheel efficiencies, fuel versatility, infrastructure, availability, economics, regeneration, and safety. 

Hydrogen fuel cells for automotive applications offer unprecedented vehicle efficiencies around 60%. The shortcoming of hydrogen is the ability to store 5.6 kg of hydrogen on an automotive platform while meeting all of the U.S. Department of Energy’s (DOE) technical system targets. Onboard hydrogen storage for light duty vehicles [1] liquid hydrogen offers increased volumetric capacities but requires storage temperatures around 20 K. For the cases of liquid hydrogen and cryo-compressed hydrogen, dormancy issues, gravimetric capacities, and volumetric capacities are the critical challenges in storing neat hydrogen for automotive applications. In an attempt to remove the limitations imposed by the gravimetric and volumetric storage capacities of neat hydrogen, three material-based approaches to hydrogen storage are being researched, namely, adsorbents, metal hydrides, and chemical hydrogen storage. The successful commercialization of on-board hydrogen storage for light duty vehicles requires that all of the DOE targets for on-board hydrogen storage systems for light-duty vehicles be met simultaneously. The DOE system-level targets for on-board hydrogen storage system for light duty applications can be seen in Table 1.

The U.S. Department of Energy’s (DOE) Fuel Cell Technologies Office has funded the Hydrogen Storage Engineering Center of Excellence (HSECoE) composed of a team of universities, industrial corporations, and federal laboratories with the objective to identify and advance the engineering work scope related to on-board hydrogen storage systems. The focus of this report centers on our work related to chemical hydrogen storage. Given that there is no chemical hydrogen storage material that meets all of the requirements for commercialization, the two leading candidates, alane (aluminum hydride, AlH_3_) and ammonia borane (NH_3_BH_3_, AB), were used as surrogates for component development, testing, and validation and also for system design and modeling. Component and system designs were later used to develop material property guidelines for chemical hydrogen storage materials expected to meet the DOE technical targets. Chemical hydrogen storage materials are sought after because of the high gravimetric and volumetric capacities and low pressure–moderate temperature operation. The primary disadvantage of chemical hydrogen storage is the off-board regeneration of spent fuel. The regeneration of chemical hydrogen storage was outside the HSECoE work scope. Our research focus was to identify, develop, and advance the shortcomings associated with the current state-of-the art chemical hydrogen materials, alane and ammonia borane. Notable shortcomings include material properties, single- and multi-phase transport, purification, reactivity, and selectivity. This report details our research findings related to chemical hydrogen storage not only in the narrow context related to ammonia borane and alane but also in the broader, more general context of chemical hydrogen storage materials. This manuscript complements the published works [2,3,4,5,6,7,8,9] that put into perspective the challenges of hydrogen storage for automotive applications.

## 2. Materials Operating Requirements

As part of the Hydrogen Storage Engineering Center of Excellence (HSECoE), the Materials Operating Requirements (MOR) Technical Area provided measured data of the materials engineering properties to support the systems modeling effort and other engineering efforts within HSECoE and to select chemical hydrogen storage (CHS) to demonstrate sub-scale systems. 

The materials engineering properties include hydrogen capacity, kinetics, thermodynamics, densities of reactants and products, form factor, viscosity, settling/flocculation, thermal stability, and thermal conductivity. We selected candidate materials for the materials groups, i.e., metal hydrides, chemical hydrides, and adsorbent materials, and provided materials engineering properties to fill in gaps in the literature and to validate the systems modeling.

The Materials Operating Requirements (MOR) were established with the goal to optimize performance of a chemical hydrogen storage (CHS) material to meet the MOR based on the evaluation of systems models. The CHS material was selected based on materials selection criteria that were established by the HSECoE MOR group described in Section 2.1. The experimental data were used to validate the systems models for design of the sub-scale system. The literature and measured data were used in the systems modeling in an iterative process between experiments and modeling to validate and refine the systems models.

### 2.1. Materials Selection Criteria

A guideline for how to justify the selection of HSECoE materials was developed. In order for a material to be considered for application within the HSECoE, it must first pass minimum screening criteria which gave the Center a rough assessment of its capabilities. The quantified minimum screening criteria for each materials group, i.e., metal hydrides, chemical hydrides and adsorbent materials, are listed in Table 2.

A material that passes the screening criteria was selected for further consideration within the HSECoE. Materials not found to improve system performance relative to selected materials and that cannot meet the DOE system targets were not for further consideration.

The selected materials were grouped into selected (Tier 1 and 2) and not selected materials according to Table 3. For Tier 1 “Developed Materials”, a database was assembled and appropriate parametric models developed and verified where system data were available. Tier 2 “Developing Materials” were selected as promising candidate materials for system consideration. The “Not furthered considered” were not found to improve system performance relative to selected materials, and thus were not for further consideration.

A materials selection procedure based on three steps was developed and is illustrated in Figure 1.

Step 1: Pass minimum screening criteria to select materials for further consideration or to be removed from further consideration.Step 2: Selected Tier 2 material is under performance evaluation and materials properties are collected.Step 3: Selected Tier 1 material undergoes system analysis and engineering properties are measured.

The Decision to Not Consider a Material for Engineering Can be Made:After comparing the materials’ engineering data with the “screening criteria” for a candidate material and finding that the candidate does not meet the minimum criteriaFor a Tier 2 material, after evaluating the collected/measured data for system considerations, it may be revealed that it will not improve performance compared to Tier 1 materials.For a Tier 1 material, during system modeling, it may be revealed that the material will not improve performance and will not meet the DOE system targets.

The Decision to Continue Considering a Material can be Made:After comparing the materials’ engineering data with the “screening criteria” for a candidate material and finding that the candidate meets the minimum criteria. Thus, move the material to Tier 2.For a Tier 2 material, after evaluating the collected/measured data for system considerations, it may be revealed that it will improve performance compared to Tier 1 materials. Thus, move the material to Tier 1.For a Tier 1 material, after performing system modeling, it may be revealed that the material will improve performance and meet the DOE system targets. Thus, it will remain in Tier 1.

### 2.2. Selection of Chemical Hydrogen Storage Material

The materials considered for CHS were ammonia borane (NH_3_BH_3_, denoted as AB), lithium alanate, LiAlH_4_ and aluminum hydride (AlH_3_, denoted as alane). Each was evaluated against the established MOR, resulting in selecting solid AB as a baseline material for further development. Upon recommendation of the Chemical Hydride Center of Excellence (CHCoE), solid AB was at an early stage selected as a baseline material because it meets the “Minimum screening criteria” for chemical hydrides and showed promise to also meet the DOE targets. It was selected to Tier 1 and system modeling was initiated.

Although it meets most of the 4:40 criteria, there are two hurdles that cannot be resolved if using solid AB “as is”, i.e., excessive foaming upon hydrogen release and generation of volatile impurities which would be detrimental for the fuel cell. Therefore, a decision was made to select solid AB “as is” and move forward with composite materials by adding methyl cellulose (MC) to AB, referred to as AB/MC, as a Tier 2 developing material. AB/MC released hydrogen at ~20 °C lower than neat AB and at a faster release rate at 160–300 °C [7]. The decision to not proceed with neat solid-phase AB was made after recommendations by the CHCoE.

### 2.3. Materials Selection Criteria for Fluid Chemical Hydrogen Storage

The HSECoE team used the materials operating requirement (MOR) to select which chemical hydrogen storage to move forward with for large-scale reactor testing. Two candidates were evaluated, Liquid AB and Slurry AB, by comparing the quantified properties in Table 4. It became evident that both materials have mostly favorable properties, meeting the minimum criteria. Considering that we needed to prepare larger quantities for scale-up experiments, we concluded that slurry AB was more advanced and we could make larger quantities for reactor testing fairly rapidly. In conclusion, we decided to move forward with Slurry AB for larger scale testing.

## 3. Alane Material Properties

Ammonia Borane (AB, NH_3_BH_3_), aluminum hydride (Alane, AlH_3_), and related compounds, such as LiAlH_4_ and Mg(AlH_4_)_2_, offer a fundamentally different approach to solid state hydrogen storage whereby hydrogen is trapped in a metastable solid state. These materials offer extremely high hydrogen densities at ambient pressure and temperature (>2× liquid hydrogen density), with a low hydrogen release temperature (<150 °C) making them useful for on-board hydrogen storage. In addition, they can supply extremely high hydrogen pressures (>1 kbar) at moderate temperatures (<150 °C), making them a useful source of high-pressure hydrogen without the need for a compressor. However, the utilization of these metastable materials in a hydrogen storage system introduces two important new challenges: (i) controlling the rate of hydrogen release in a system far from equilibrium and (ii) regenerating or reforming a hydride from the hydrogen-depleted material (i.e., hydrogenating a material with a very low formation enthalpy).

### 3.1. Alane Synthesis and Structures

Aluminum hydride is an attractive (endothermic) metastable hydride due to its high gravimetric density (10.1 wt%H) and volumetric hydrogen density (148 g H_2_/L), its low reaction enthalpy (7 kJ/mol H_2_), and its ability to release hydrogen at rapid rates at low temperature (<150 °C). In addition, AlH_3_ decomposes to Al and H_2_ in a single step with no side reactions or undesirable intermediates.

Aluminum hydride is typically prepared via a reaction between LiAlH_4_ and AlCl_3_ in diethyl ether, which forms an alane etherate adduct [10,11]. Other tetrahydroaluminates or binary hydrides and other aluminium salts can also be use in this process. The solution process results in the precipitation of a salt byproduct (e.g., LiCl), which is removed by filtration. The etherate solution is then desolvated under dynamic vacuum at 65–75 °C to recover crystalline AlH_3_. Typically, excess LiAlH_4_ is used along with LiBH_4_, which has been shown to reduce the time and temperature of desolvation. The excess hydrides are removed by a wash in diethyl ether after desolvation. The direct desolvation of the ethereal solution (drying under dynamic vacuum) yields irregular sub-micron (200–500 nm) aluminum hydride [12]. A continuous process developed by Brower et al. can be used to prepare larger quantities of AlH_3_ with larger cubic crystallite sizes (50–100 μm) [12]. In this steady state process, the desolvation is performed in toluene, and the AlH_3_ etherate solution is added continuously at ~77 °C [10,11]. Larger crystals may be washed in an aqueous solution (e.g., dilute HCl) without complete hydrolysis. This wash removes impurities and passivates the hydride surface to improve shelf life and can be used to tune the dehydrogenation temperature.

Aluminum hydride forms a number of different crystalline polymorphs, including α, α’, β, γ, δ, ε, and ζ. The most stable and by far the most commonly used polymorph is α-AlH_3_, which is composed of corner connected AlH_6_ octahedra bridged by H atoms (trigonal space group, *R*$\bar{3}$*c*). The other polymorphs have similar structures with corner or edge sharing octahedra bridged by H atoms [10]. The less stable polymorphs are common impurities in the synthesis of α-AlH_3_ and are generally considered undesirable due to their propensity to decompose (dehydrogenate) at room temperature.

### 3.2. Alane Thermodynamics and Kinetics

Dehydrogenation of α-AlH_3_ to H_2_ and Al is entropically driven (as with most hydrides) and occurs in a single endothermic step (α-AlH_3_ → Al + 3/2H_2_) with a reaction enthalpy of ~7 kJ/mol H_2_ and an entropy of ~129 J/K mol H_2_ giving a reaction free energy of ~−31 kJ/mol H_2_ at 300 K [13]. The other polymorphs are less stable than α-AlH_3_ by 1–2 kJ/mol H_2_ with a similar entropy. α-AlH_3_ has an equilibrium H_2_ pressure of 1 bar at −218 °C, however H_2_ evolution from α-AlH_3_ is kinetically limited up to temperatures of ~100 °C. Hydrogenation of aluminum metal (to form AlH_3_) is exothermic, but requires a high hydrogen pressure (~7 kbar at room temperature) to overcome the entropic barrier to forming the hydride [10].

The thermal desorption kinetics of α-AlH_3_ are typical for solid-state hydrides. The rate of hydrogen release has an Arrhenius relationship with temperature, but is also dependent upon composition (value of *x* in *x*Al + (1 − *x*)AlH_3_). The isothermal desorption curve (fractional decomposition vs. time) has a sigmoidal shape (Figure 2a) consisting of an induction period, corresponding to nucleation of the aluminum phase, an acceleratory period, attributed to growth of the metal in three dimensions, and a decay period. The less stable polymorphs of AlH_3_ typically undergo an exothermic transition to the α phase at elevated temperatures (≥100 °C). Although these transitions provide a small amount of heat to the material, no H_2_ is evolved during this transition, and therefore the isothermal decomposition kinetics for α, β and γ-AlH_3_ are similar above 100 °C [10,12]. In contrast, at lower temperatures the decomposition of the less stable polymorphs is more complex with a fraction of the material decomposing directly to the elements (e.g., α-AlH_3_ → Al + 3/2H_2_), while the remaining material transforms to α-AlH_3_ before decomposing (e.g., α-AlH_3_ → α-AlH_3_ → Al + 3/2H_2_) [10,12]. Figure 2b shows the rates of the α, β and α polymorphs, showing faster rates for the less stable polymorphs (β and γ) at <100 °C and similar rates for all three polymorphs above 100 °C.

In a practical system, the use of AlH_3_ or another metastable hydride requires a mechanism to control the rate of hydrogen release so that it is compatible with the demand from the fuel cell. Most reversible hydrides (e.g., FeTiH_x_ or MgH_2_) operate near equilibrium, and therefore the hydrogen pressure is managed by maintaining a constant tank temperature. As hydrogen is consumed by the fuel cell, the material replenishes the lost hydrogen up to the equilibrium pressure and will maintain that pressure as long as there is hydrogen left in the material. A metastable hydride operates far from equilibrium, and once a suitable activation energy is supplied, hydrogen will continuously evolve from the material, raising the pressure with no equilibrium (nearby) to stop it. There are a number of possibilities to control the rate of hydrogen release including zoned or partitioned tanks, where individual heaters are used to fully decompose an aliquot of hydride as needed. Similarly, a more continuous version of this concept involves flowable hydride systems, where the hydride is pumped into a reactor zone and decomposed. In this case, the H_2_ rate is controlled by the amount of material moving through the reactor. Another possibility for controlling hydrogen evolution is to establish a rate equation (or lookup table) that relates the hydrogen release rate to the temperature and composition of the hydride. When this relationship is incorporated into a simple electronic circuit capable of controlling the tank temperature, a feedback can be established which converts a drop in pressure (or demand from the fuel cell) into a temperature change and a hydrogen flow rate. This system is static (no hydride transport) and only requires a single heater, and therefore could be engineered into a very low cost (i.e., disposable) canister. An example of this type of hydride canister was recently demonstrated with a proton exchange membrane (PEM) fuel cell showing continuous power generation for ~7 h [14].

### 3.3. Alane Regeneration

One of the primary challenges associated with using a metastable hydride as a solid-state hydrogen carrier is the reformation of the hydride from the spent fuel and hydrogen gas. In most cases, direct re-hydrogenation is not possible or requires extremely high pressures. Widespread use of alane (or other metastable hydrides) requires the development of a low-cost recycling or regeneration process to chemically re-form the hydride from the spent fuel and H_2_ gas. One obvious approach is to simply recycle the byproducts and spent fuel to recover the precursors used in the conventional synthesis process. However, in this case the energy required to overcome the thermodynamic barrier to reduce the salt byproduct is high, leading to a maximum (theoretical) regeneration efficiency of 30%, far short of the well-to-tank efficiency target of 60% [10].

Another approach to re-forming alane involves the electrochemical hydrogenation of aluminum at low pressures. Direct electrochemical hydrogenation of aluminum is difficult, and although some hydrogen incorporation into Al is possible at low pressures (<1 kbar), the H concentrations are low (~1000 ppm) with no formation of α-AlH_3_ [15]. More recently, Zidan et al. have demonstrated the feasibility of electrochemically charging Al with H using an Al anode and an electrolyte of NaAlH_4_ in tetrahydrofuran (THF) to form an alane adduct (AlH_3_-THF) [16,17]. In this reaction, NaAlH_4_ is used as a hydrogen source via AlH_4_^−^ ions, which combine with Al from the anode to generate solvated AlH_3_ at the anode (3AlH_4_^−^ + Al → 4AlH_3_·*n*THF + 3e^−^). This is similar to the conventional chemical synthesis reaction where the Al^3+^ ions from AlCl_3_ are replaced by Al^3+^ from the anode. A typical electrochemical reaction forms Na metal or Na_3_AlH_6_ (formed from Na+ and NaAlH_4_) at the cathode which can introduce a number of challenges such as growth of dendrites (cathodic product) at high current densities and the recovery and conversion of the cathodic product back to hydride [17]

The formation of alane complexes (e.g., alane amine) can also be accomplished through a direct thermochemical (rather than electrochemical) route. Hydrogenation of catalyzed Al and an amine (NR_3_) can occur at low pressures in a liquid (slurry) medium to form an amine alane (AlH_3_·NR_3_). A number of different tertiary amines are suitable, including trimethylamine (TMA), triethylenediamine (TEDA), dimethylethylamine (DMEA), and others [10,18] The thermochemical hydrogenation is more direct route to forming the alane complex (presumably it can be used directly with the Al powder from the spent fuel), but may also be more limited in the types of adducts that can be formed.

The primary obstacle with both approaches is the separation of the alane adduct to recover α-AlH_3_. Similar to the conventional synthesis, the ligand is removed from the alane via a thermally driven process (typically under vacuum), which needs to be performed with some care to avoid decomposing the hydride. In general, the more stable alane adducts (e.g., AlH_3_·TEDA) that form readily under mild hydrogenation conditions are the most difficult to separate. Conversely, alane adducts that are easily separated, such as alane diethyl ether (AlH_3_·Et_2_O) or alane triethylamine (AlH_3_·TEA), have proven difficult to form by direct hydrogenation. To overcome this issue, a transamination or ligand exchange step can be employed to exchange the amine with one that forms a less stable adduct. An example of a direct regeneration path for the formation of AlH_3_ (showing the calculated structures of each component) is shown in Figure 3 which employs the hydrogenation of DMEA and Al to form AlH_3_·DMEA, followed by incorporation of TEA and transamination to AlH_3_·TEA, followed by adduct separation to recover AlH_3_ [10]. A similar regeneration approach could be employed for other metastable hydrides such as LiAlH_4_ and Mg(AlH_4_)_2_.

### 3.4. Alane Outlook and Challenges

The high gravimetric/volumetric hydrogen density and low H_2_ evolution temperature (requiring little to no heat input) characteristic of the metastable hydrides makes them interesting candidates for portable power systems for military (soldier power) and commercial (wearable devices, backpacks, jackets, and clothing) applications. Other near-term applications include unmanned aerial and underwater systems. A metastable hydride, such as AlH_3_, coupled with a lightweight fuel cell has a projected useful specific energy >800 Wh/kg, nearly four times greater than a state-of-the-art lithium ion battery pack. This implies that an all-electric unmanned aerial system would have 4× the flight time of a comparable battery-powered system. The growing interest in hybrid-electric and all electric passenger and cargo planes suggests a possible role for metastable hydrogen carriers in larger systems as well. The high specific energy combined with the conformable nature of the tank and the ability to “refuel” as opposed to “recharge” makes a metastable hydride fuel cell system an interesting option for some of the larger scale mobile platforms. Other applications include the use of metastable hydrides for the transportation and generation of high-pressure hydrogen. Metastable hydrides can store and transport hydrogen at densities greater than those of high pressure or cryogenic storage systems, but at an ambient pressure and temperature. In addition, these materials can be used to generate extremely high pressures (>1 kbar) without the need for a compressor or high temperatures.

Despite these interesting opportunities for metastable hydrides, the commercialization of a metastable hydride fuel cell will be highly dependent upon the cost of the fuel. This suggests the need for new, low-cost synthesis procedures (increasing yield and reducing energy requirements) and regeneration routes to reform the hydride directly from the spent fuel. Some possible pathways for reducing the cost could include lowering or eliminating solvents from the conventional synthesis reaction, identifying methods to form aluminum hydride adducts directly from aluminum metal (as opposed to dehydrogenated aluminum hydride), and low energy, high yield methods of separating aluminum hydride adducts to recover the hydride.

## 4. Ammonia Borane Material Properties

### 4.1. Ammonia Borane Synthesis and Structures

Ammonia borane (NH_3_BH_3_, denoted as AB) is a solid-state hydrogen storage material that has one of the highest known net-usable gravimetric hydrogen capacities around 17–17.6 wt% (g_H2_/g_AB_). Ammonia borane was first synthesized by Shore via the metathesis of lithium borohydride and ammonium chloride in diethyl ether [19]. The general reaction scheme can be represented by the following reactions.
$${\mathrm{LiBH}}_{4}+{\mathrm{NH}}_{4}\mathrm{Cl}\to {\left[{\mathrm{NH}}_{4}\right]}^{+}{\left[{\mathrm{BH}}_{4}\right]}^{-}+\mathrm{LiCl}$$
$$\frac{{\left[{\mathrm{NH}}_{4}\right]}^{+}{\left[{\mathrm{BH}}_{4}\right]}^{-}\to {\mathrm{NH}}_{3}{\mathrm{BH}}_{3}+H_{2}}{{\mathrm{LiBH}}_{4}+{\mathrm{NH}}_{4}\mathrm{Cl}\to {\mathrm{NH}}_{3}{\mathrm{BH}}_{3}+H_{2}+\mathrm{LiCl}}$$

Ammonia borane is a colorless waxy solid with having a molecular weight of 30.87 g/mol, a density of 0.78 g/mL, and a melting point of around 104 °C. Ammonia borane has an ultimate hydrogen capacity of 19.4 wt% when AB is fully decomposed to boron nitride. Ammonia borane was extensively studied in the DOE-funded Chemical Hydrogen Storage Center of Excellence (CHSCoE) as a hydrogen storage material for automotive fuel cell applications [3]. Of the many chemical hydrogen storage materials studied in the CHSCoE, ammonia borane and alane were down-selected for further study-primarily based on their volumetric and gravimetric hydrogen capacities and the likelihood in meeting the DOE system level targets. For a thorough review of ammonia borane and its properties, the reader is directed to the following references [2,3,6], in particular, the review by [4].

### 4.2. Ammonia Borane Thermodynamics and Kinetics

The thermal decomposition (also referred to as thermolysis) of solid-phase ammonia borane to produce hydrogen occurs through a series of exothermic reactions. The onset temperature for AB decomposition is around 85–90 °C and continues to release hydrogen for temperatures well in excess of 500 °C. The ultimate thermodynamically stable product is boron nitride; however, for the purposes of using ammonia borane as a hydrogen storage medium that can be regenerated, boron nitride is not the sought-after product. Boron nitride is a thermodynamically stable, highly refractory ceramic that is for all practical purposes not thermodynamically feasible to regenerate back to ammonia borane. Therefore, the amount of hydrogen produced from the thermal decomposition of ammonia borane should be restricted to less than 2.3 moles of hydrogen per mole of AB [3]. The decomposition enthalpy of ammonia borane for the production of 2–2.5 equivalents of hydrogen is on the order of 37–43 kJ/mol of ammonia borane [20,21]. The common products observed during AB decomposition include linear, branched, and cyclic BNH compounds (refer to Figure 4).

The thermal decomposition of ammonia has been extensively studied in the neat solid-phase (with and without additives), solution-phase, and slurry-phase. Representative kinetics for neat solid-phase ammonia borane are shown in Figure 5 for the isothermal temperatures in the range of 160–300 °C [3,7]. Typically observed with the thermal decomposition of ammonia borage is an induction period prior to the onset of hydrogen release. As expected, the rate of hydrogen production increases with increasing temperature with a maximum hydrogen release of 2–2.5 equivalents of hydrogen after 60 min.

Although ammonia borane is a metastable hydrogen carrier, the kinetics of hydrogen release (including the induction period) at 300 °C are still too slow for the demanding drive cycles for automotive fuel cell applications. Himmelberger et al. was the first to demonstrate increased kinetic activity for ammonia borane dehydrogenation using ionic liquids [22]. Shown in Figure 6 are the isothermal (T = 85 °C) dehydrogenation kinetics ofa 50:50 mixture of ammonia borane-BmimCl (1-Butyl-3-methylimidazolium chloride) composition. Ammonia borane compositions with BmimCl (1-Butyl-3-methylimidazolium chloride) demonstrated an increase in not only hydrogen evolution kinetics but also the overall hydrogen yield as compared to neat solid-phase ammonia borane. The dehydrogenation kinetics of a 50:50 AB:BmimCl were also investigated as a function of temperature and are shown in Figure 7. The kinetics of dehydrogenation increased with increasing temperature with the highest hydrogen release rate occurring at the highest temperature investigated (110 °C). It is important to note that the addition of BmimCl removed the induction period typically observed with neat ammonia borane. 

Owing to the difficulties of implementing an engineered hydrogen storage and delivery system for automotive fuel cell applications for hydrogen storage materials requiring off-board regeneration, liquid-phase hydrogen storage materials are likely the only viable path forward. Liquid-phase hydrogen storage media can be neat liquids, solutions or slurry-phase. The challenges associated with liquid-phase compositions include (to name a few) solubility, freezing points, boiling points, thermal stabilities of solvents or carriers (for slurries), cost, system complexity, and regeneration chemistry [2]. The material properties of the liquid-phase hydrogen storage media go well beyond just dehydrogenation kinetics, gravimetric capacity, volumetric capacity, and cost. Thermodynamic solubility of ammonia borane in polar solvents is being actively researched with the goal of identifying a suitable solvent for ammonia borane dissolution [23].

### 4.3. Ammonia Borane Regeneration

The efficient and cost-effective regeneration of ammonia borane was a focus of the CHSCoE because without an efficient and cost effective regeneration process the global implementation of ammonia borane as a hydrogen storage material is highly unlikely. The DOE cost target for delivered hydrogen is USD 2–3 per kg of hydrogen at the pump (Table 1). This implies that the cost of regeneration must be lower the USD 3/kg H_2_ to meet the DOE target. Various approaches were investigated in the regeneration spent ammonia borane (i.e., polyborazylene), namely, the thiacatechol approach, the alcohol approach, the super acidic haloacid approach, and the hydrazine approach [3,5,24,25]. A complete review of these approaches can be found in the summary report from Dow [5]. The hydrazine approach is the most elegant of the regeneration approaches in that it is a one-pot process that directly converts polyborazylene (spent AB) to ammonia borane in the presence of ammonia and hydrazine at 60 °C [24]. The general reaction scheme can be represented as follows:$${\mathrm{BNH}}_{2}+N_{2}H_{4}\overset{{\mathrm{NH}}_{3}}{\to }{\mathrm{NH}}_{3}{\mathrm{BH}}_{3}+N_{2}$$
ΔHrxn, 25 °C=−6.95 kcalmol
ΔGrxn, 25 °C=−22.88 kcalmol

The cost of regenerating spent ammonia borane with the hydrazine process is USD 45.73 per kilogram of hydrogen, with the primary cost being hydrazine at USD 43.79 per kilogram of hydrogen [5]. Although the hydrazine one-pot process remains the most elegant and facile process to regenerate spent ammonia borane, the cost of regeneration and the impacts on the overall cost of hydrogen delivered at the pump cannot be overlooked by material researchers in their quest for hydrogen storage materials.

### 4.4. Ammonia Borane Outlook and Challenges

Ammonia borane has an attractive net-usable hydrogen capacity of around 2.5 moles of hydrogen per mole of ammonia borane (~17 wt% H_2_), relatively low thermal decomposition temperatures (<200 °C), and moderate kinetics. However, the in the grand view of using ammonia borane as a hydrogen storage material for automotive applications, the regeneration cost and the engineering nuisances that are often overlooked by materials researchers (solid-phase transport, cost, off-board regeneration, etc.) will prevent ammonia borane from being commercially implemented for this application. Ammonia borane may find its way in niche applications that do not require regeneration (e.g., drones, portable power), analogous to alkaline batteries.

## 5. Liquid Phase Dehydrogenation Reactors with Kinetics

### 5.1. Objective

Methoxypropyl amine borane (MPAB) was originally developed as hydrogen bearing solvent (3.9 wt% H_2_) for the dissolution of ammonia borane [26,27]. Given the relatively high hydrogen content, high boiling point, and, most importantly, no phase change upon dehydrogenation, MPAB was used as the liquid-phase chemical hydrogen storage surrogate for designing and demonstrating liquid-phase dehydrogenation reactors. Mixtures of ammonia borane and MPAB resulted in an unstable composition that released hydrogen upon mixing at room temperature. Due to the room temperature instability of AB/MPAB compositions, they were discontinued from further research. Although MPAB does not contain the necessary gravimetric capacity required for automotive applications, MPAB did however offer the opportunity to investigate a hydrogen-bearing neat liquid that does not undergo a phase change upon dehydrogenation. The dehydrogenation reaction of MPAB can be written as



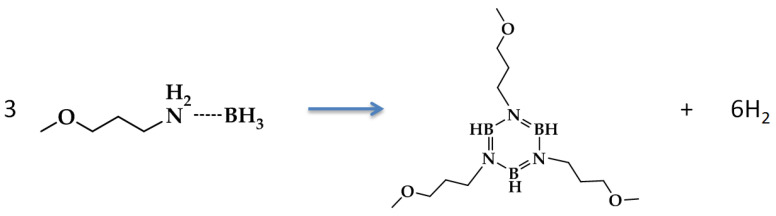



The salient features of MPAB dehydrogenation reaction include the large volume of hydrogen generated and the decrease in liquid molar flow rate. Even with a low gravimetric capacity of 3.9 wt% H_2_, the ratio of reactant liquid flow rate to the volumetric flow rate of hydrogen at 100% conversion is substantial (at STP, the ratio is ~443 L H_2_/L MPAB); the ratio is even higher when referenced to the product flow rate (i.e., ~1330 L H_2_/L product). The ratio of volumetric flow rates of hydrogen generated to the liquid-phase reactant (or products) will be exacerbated for media possessing hydrogen gravimetric capacities much greater than 3.9 wt% H_2_. The large flow rate ratio poses reactor design issues when attempting to build a reactor with the smallest form factor and volume while also maintaining high conversions. The primary technical barriers or challenges with the design and operation of reactors using liquid-phase chemical hydrogen storage media include the following:

### 5.2. Technical Barriers and Challenges

Reactor slugging due to hydrogen gas generation;Freezing and boiling point of liquid-phase chemical hydrogen storage media;Phase change post-dehydrogenation;Dehydrogenation kinetics and reactor operating temperature.

The technical barriers or challenges listed above contribute to an overall decrease in reactor performance. Reactant slugging is caused by the rapid evolution of gas (i.e., hydrogen) that expels liquid reactant from the reactor faster than otherwise desired or expected; in other words, the space–time is decreasing. This is undesirable because of the low single-pass conversions (and potentially limited overall conversions) observed. Although we did not perform reactor tests below the freezing point of MPAB, we do expect problematic operation with the pumping and reactor performance. Inline heaters and tank heaters are likely to be implemented if such problems are encountered. Boiling point (and vapor pressure) is a serious concern if the dehydrogenation kinetics are too slow, thereby requiring reactor temperatures approaching the boiling point of the liquid or a larger reactor. Reaction types other than unimolecular decomposition will be burdened by volatile reactants because of an increase in mean free-path between reactants, a decrease in space–time, and also because of the decrease in gas-phase reactant concentration, primarily attributed to the hydrogen produced. Any significant reactant vapor pressure will cause unreacted reactants to be transported downstream of the reactor, thus requiring additional component mass and volume for capture and recycle. Phase change upon dehydrogenation (specifically liquid to solid), is undesirable. Dehydrogenation kinetics and reactor temperature are closely coupled. If the kinetics are too slow, then the reactor temperature must be increased, or catalysts must be developed to accommodate the tight reactor constraints of mass and volume. Increasing the reactor temperature may have the unintended consequence of increasing the partial pressure of the reactant and thus decreasing the single pass conversion as described above. If the kinetics are too fast, then the material may be unstable at room temperature (e.g., mixtures of AB and MPAB). In short, there is a narrow window of intersection between kinetics and thermodynamics for materials meeting the DOE 2017 technical targets.

### 5.3. Approach

Our approach in designing reactors for liquid-phase chemical hydrogen storage media focused primarily on reactor slugging. The other technical barriers are material specific properties and we did not attempt to change these with catalysts and/or additives. We focused on addressing the engineering challenge of designing the most efficient reactor with the smallest footprint. Given the small volumetric flow rate of liquid relative to the volumetric flow rate of hydrogen (at 100% conversion), we settled on two primary approaches to mitigate reactor slugging, namely, helical and packed bed reactors. In addition to mitigating reactor slugging, these approaches also offer the ability to perform preliminary gas–liquid separations thus reducing the burden of the primary downstream gas–liquid separator. Pictorial representations of both the helical reactor and the packed-bed reactor are shown in Figure 8. Note that numerous design variations of the helical reactor can be expected by varying the thread pitch, thread depth, and thread angle.

The two reactors were integrated into a prototype test system designed to mimic the primary components envisaged to be present in a realizable automotive hydrogen storage and delivery system. The primary components included heat exchangers, gas–liquid separator, hydrogen purification, and a dehydrogenation reactor. The analytical instrumentation employed to quantify reactor performance included gas-phase infrared spectroscopy and gas chromatography. The quantified species were borazine, hydrogen, ammonia, and diborane. The bench-scale system used for assessing the reactor performance is shown in Figure 9.

### 5.4. Summary of Findings

In general, the helical reactor designs mitigated reactor slugging and worked equally well in the downward, upward, or horizontal flow directions. With the aid of high volumetric flow rates of hydrogen, the helical reactor design facilitated the outward or radial flow of liquid reactants and products toward the reactor tube wall. Maintaining the liquid-phase reactants and products on the reactor wall promoted gas–liquid separation, prevented reactor slugging, and also promoted oligimerization of MPAB by maintaining a continuous liquid film. The packed-bed reactor also mitigated reactor slugging by providing a highly tortuous, high surface area packing that prevented aerosol droplets from facile straight-through transport within the reactor. The packed-bed reactor had on average 16% higher hydrogen space–time yields than the helical reactors. Although the packed-bed reactors outperformed the helical reactors, packed-bed reactors will likely be limited to either the downward or upward flow orientation, probably more suited for stationary applications. The helical reactors were observed to work equally well in the downward, upward, and horizontal flow orientations, thus implying a reactor performance that is invariant to reactor orientation. This is promising for vehicular applications because of the ever-changing orientations encountered during real-world driving. The shortcoming encountered with MPAB for all reactors tested was the increased partial pressure at elevated reactor temperatures (>200 °C) that prevented us from observing single-pass conversions greater than 70% (Figure 10A,B). Analysis of the liquid-phase products via NMR indicated MPAB, borazanes, and a borazine derivative (Figure 10C). Gas-phase products were primarily limited to hydrogen. Borazine, diborane, and ammonia were essentially baseline (Figure 10D). It is assumed that vapor-phase MPAB substantially decreased the rates of reaction (and hence limited the single-pass conversion) due to a decrease in the reactor residence time, an increase in mean-free path for a trimolecular reaction (or two bimolecular reactions in series), and lastly, an increase in the gas-phase concentration of MPAB. Consequently, larger reactors operating at lower temperatures or multiple-pass reactor operation are likely to be employed for chemical hydrogen storage media exhibiting limited single-pass conversions due to feed vaporization.

### 5.5. Remaining Challenges and Opportunities

The remaining challenges are on the development of a liquid-phase chemical hydrogen storage material whose dehydrogenation kinetics are fast enough to achieve nearly 100% conversion at reactor temperatures far below the boiling point of the storage media. Ideally, the chemical hydrogen storage material will remain a liquid both pre- and post-dehydrogenation. Material properties such as viscosity, boiling point, and freezing point will also prove to be critical in the design, implementation, and durability of the overall system design. If the partial pressures of the hydrogenated fuel or dehydrogenated fuel are too high, then recycle reactors coupled with additional purification/separation schemes are likely to be implemented, adding at a minimum mass and volume to the overall system. The novel helical reactor performed as designed by promoting gas–liquid separation and preventing reactant slugging thus maintaining the necessary space–time of the liquid-phase reactants. This design optimized reactant conversion while minimizing the reactor footprint regarding mass and volume. This design is seen as a breakthrough in reactor design for reactions that generate large volumes of vapor-phase products. The helical reactor design is also expected to outperform traditional reactors involving both gas and liquid-phase reactants.

## 6. Slurry-Phase Dehydrogenation Reactors with Kinetics

### 6.1. Objective

The objective of this study was to use alane and AB slurries (20–65 wt% loadings) as surrogates to develop and demonstrate the feasibility of flow-through dehydrogenation reactors using slurry-phase chemical hydrogen storage media.

### 6.2. Technical Barriers and Challenges

Fouling/clogging due to slurry stratification;Reactor slugging due to hydrogen gas generation;Three-phase flow;Chemical compatibility between solid and liquid phases;Freezing/boiling points of liquid phase;Dehydrogenation kinetics;Mixing components for homogeneously dispersed slurry compositions.

### 6.3. Approach

Our approach to designing slurry-phase reactors focused on addressing reactor fouling and reactor slugging. A way of addressing reactor fouling is through the mechanically mediated reactive transport of slurry compositions. A motor driven auger was used to mechanically facilitate solids transport (i.e., AB, spent AB, Alane, and Al) through the reactor in expectation of preventing reactor fouling/clogging. The auger design is analogous to a ship auger bit which allowed for gas-phase transport through the centerline of the auger (see Figure 11). Mechanical mixing also served to enhance gas separation of entrained hydrogen and to prevent solids agglomeration. We used a Watson–Marlow peristaltic for feed delivery (gear pumps were tried but seized due to clogging). We did not use syringe pumps because we required homogenous feed mixtures for correlating observed kinetics. All slurry feed reservoirs required continuous mixing with a motor driven impeller to maintain feed homogeneity. Depending on the slurry loading, phase separation could be observed in as little as 30 min. Numerous design iterations to the testing system were performed to address the unanticipated non-idealities encountered with ammonia borane slurries. The general schematic of the last design iteration of the testing system can be seen in Figure 12. Similar to the testing system used for MPAB (refer to Figure 9), the slurry-phase testing system contained heat exchangers, gas–liquid separators (primary and secondary), and a dehydrogenation reactor. The primary differences include a feed reservoir mixer, flow conditioning, diborane conditioning, and particulate filter. With the exception of the feed reservoir mixer, the added components were specifically implemented to address the non-idealities of AB slurries. The analytical equipment used for gas-phase analyses with slurry-phase media includes gas-phase IR spectroscopy and gas chromatography. The quantified analytes were borazine, hydrogen, ammonia, and diborane. Alane (as-received from ATK) was prepared in a slurry with silicone oil (CAS #: 63148−58−3), mechanical pump fluid (CAS #: 64742−65−0), and tetraglyme (CAS #: 143−24−8). Alane slurry loadings ranged from 20–65 wt% solids loadings. Lithium hydride (CAS #: 7580−67−8) and titanium t-butoxide (CAS #: 3087−39−6) were also used as alane slurry additives to promote dehydrogenation. Ammonia borane slurries were prepared with silicone oil (CAS #: 63148−58−3) and Triton X−15 (anti-foaming surfactant, CAS #: 9036−19−5) ranging from 20–50 wt% solids loadings. Choi and Westman investigated preparation methods and optimized the AB slurry to reach up to 50 wt% solid loadings in the silicone oil by reducing particle size and selecting tip-sonication for mixing which was highly beneficial for obtaining well-dispersed slurries and acceptable flowability [8,9].

Cautionary Notes: All researchers are urged to take the necessary safety and operational precautions when performing research with chemical hydrogen storage materials. All researchers should be mindful of the following when conducting research:Safety and engineering controls (e.g., proper fire extinguishing media, pressure relief, venting, gas conditioning, proper materials of construction, etc.);Flammability/dust explosions/pyrophoric hazards (e.g., diborane, hydrogen, aluminum);Nanoparticle hazards (particles < 100 nm);Toxicity and carcinogenic hazards (e.g., diborane).Be mindful of the materials being researched and their potential products, for example,Chemical compatibility;Storage and shelf-life;Clean up and disposal.

### 6.4. Summary

Flow-through slurry-phase reactor experiments designed to collect dehydrogenation kinetics and to demonstrate the feasibility of the mechanically mediated reactive transport of slurry-phase media proved to be difficult. The difficulties of note were reactor clogging/fouling (observed for both alane and AB slurries) and the clogging of ancillary downstream components such as valves and rotameters (observed with AB slurries). The key research findings are:

#### 6.4.1. Alane slurries:

Alane dehydrogenation kinetics.Alane conversion data were limited to temperatures below 200 °C because of the complete vaporization of the tetraglyme carrier and the partial vaporization of silicone oil and mechanical pump fluid. Side reactions of alane with silicone oil and mechanical pump fluid were also observed for reactor temperatures greater than 200 °C.Complete conversion of alane was observed for a 20 wt% alane slurry in silicone oil operating at a reactor temperature of 250 °C, a space–time of 6.8 min, and an auger speed of 40 rpm. Alane conversions greater than 80% were observed for a 50 wt% alane–silicone oil slurry (T_avg_ = 205 °C, Auger = 12 rpm, space–time = 7.6 min).Lithium hydride added with titanium promotes alane dehydrogenation with nearly a doubling of the observed hydrogen mole percentage (Figure 13). Lithium hydride alone did not promote alane dehydrogenation.

#### 6.4.2. Alane Impurities

Impurity generating side reactions were observed with silicone oil-based alane slurries. The gas-phase impurities (evidenced by “Si-H” gas-phase IR transitions, Figure 14) were observed to increase with increasing reactor temperature with an onset temperature of around 200 °C. Note that alane slurries with mechanical pump fluid eliminated the production “Si-H” containing impurities. However, side reactions of alane/aluminum and mechanical pump fluid were also observed (onset temperature around 200 °C). Dehydrogenation of alane in slurry form demonstrated hydrogen selectivities less than one for reactor temperatures greater than 200 °C. Hydrogen selectivity is defined as the molar ratio of hydrogen produced to the total molar flow rate of products. In other words, a hydrogen selectivity of one indicates that the only product produced is hydrogen. Hydrogen selectivities less than one indicate that there are additional products other than hydrogen that are produced. However, alane slurries demonstrated hydrogen selectivities equal to one for temperatures less than 200 °C.

#### 6.4.3. Reactor Fouling/Clogging: Alane Slurries

Auger reactor plugged (no gas or liquid flow) within an hour of operation with 65 wt% alane/tetraglyme slurries. The failure mode was due to the complete vaporization and subsequent loss of the tetraglyme carrier.Demonstrated twelve hours of continuous operation with a 60 wt% alane/mechanical pump fluid slurry. Auger seized after twelve hours; the mode of failure was the result of solids buildup between the auger and the reactor wall.

#### 6.4.4. Ammonia Borane Slurries:

Ammonia borane impurities

Ammonia borane dehydrogenation, in both slurry and neat form, demonstrated hydrogen selectivities less than one. The primary impurities quantified include diborane, ammonia, and borazine. Side reactions of ammonia borane and silicone oil were not observed for the reactor temperatures tested (100–250 °C). Ammonia was observed to be the primary impurity (~3 mol%) for temperatures less 150 °C, while borazine was the primary impurity (followed by diborane) for temperatures greater than 150 °C. Borazine and diborane concentrations were observed to increase with increasing temperature (Figure 15). At an average reactor temperature of 167 °C, the borazine and diborane concentrations were approximately 3 mol% and 1 mol%, respectively.

#### 6.4.5. Reactor Fouling/Clogging: AB Slurries

Auger reactor tests with 50 wt% AB slurries in silicone oil were unsuccessful because the composition could not be pumped. Ammonia borane slurries proved very difficult to pump because of the “wax-like” properties of ammonia borane.Successfully demonstrated flow through reactor tests using 20 wt% AB/Si oil slurries.Plugging occurred in the ancillary components of the testing system using AB/Si oil slurries. The mode of failure was attributed to the collection of very fine powders on elbows and small orifice changes (i.e., valves and rotameters). The fine powder of the unreacted AB was observed as a “white fog” and proved very difficult to mitigate its downstream propagation. The fine powder was observed to pass through a gas–liquid separator, a silicone oil bath, a course filter, and six feet of tubing. NMR analysis of the residue collected downstream indicated fully hydrogenated and partially hydrogenated boron species [26].

#### 6.4.6. Ammonia Borane Slurry Stratification

Separation of ammonia borane and polyborazylene was observed with silicone oil-based slurries. Stratification occurred because of the differing densities of ammonia borane (0.78 g/mL) and polyborazylene (>1 g/mL) as compared to silicone oil (1.01 g/mL). Shown in Figure 16 are the stratified AB slurry compositions as a function of conversion. Unreacted or fully hydrogenated AB having a lower density than silicone oil tends to flocculate and float (far left vial in Figure 16). Partially spent or dehydrogenated ammonia borane contains both AB and polyborazylene. In this case AB flocculates and floats, while polyborazylene being denser than silicone oil tends to settle out (middle vial in Figure 16). Higher extents of ammonia borane conversion result in more polyborazylene, which leads to a greater amount of polyborazylene to settle out (far right vial in Figure 16). The density distributions observed with fresh and spent fuels will prove to be problematic in maintaining slurry homogeneity.

### 6.5. Remaining Challenges and Opportunities

The remaining challenge is to supplant the notion that solid- or slurry-phase chemical hydrogen storage materials for automotive applications requiring off-board regeneration are viable candidates. The complexities involved with solid- or slurry-phase transport in the overall fuel cycle of the storage media will have ramifications on durability, cost, efficiency, mass, and regeneration (among others), with regeneration applying not only to the chemical hydrogen storage material, but potentially to the liquid carrier as well.

## 7. Ammonia Borane Slurry Transport

### 7.1. Objective

The objective of the ammonia borane slurry transport study was to examine the ability to continuously pump a 35 wt% ammonia borane slurry through a continuous length of tubing that integrated common pipe routing (e.g., 45° and 90° bends). We also included elevations changes (10–15 inches) within the flow path to represent the elevation changes that might be expected onboard a vehicle.

### 7.2. Technical Barriers and Challenges

System durability and operability: pipe fouling/clogging

### 7.3. Approach

The approach taken for the qualitative investigations of slurry transport included a closed loop flow path with two 45° bends, four 90° bends, a reservoir, peristaltic pump, and approximately six feet of Teflon tubing (0.5′′ O.D.). Fittings were only used in transitioning to and from the peristaltic pump tubing, thus minimizing potential fouling points due to fittings. The Teflon tubing was a single piece of tubing that provided a smooth continuous flow path. As elevation changes are likely to be encountered in a vehicle flow loop, the flow loop setup also included an overall elevation change of 15 inches. The slurry used for testing purposes was a 35 wt% ammonia borane–silicone oil slurry. Slurry flow testing consisted of various flow rates ranging from 15–50 mL/min and start–stop conditions. All experiments were performed under ambient conditions. The flow loop used for testing is shown in Figure 17.

### 7.4. Summary

When 35 wt% AB/Si oil slurry solution is homogeneous and pumped through smooth surface tubing, 15–50 mL/min. (3–10 kW equivalent) pump rates can be demonstrated.When 35 wt% AB/Si oil slurry solution is homogeneous and pumped through smooth surface tubing at 50 mL/min. (10 kW equivalent), stopping the flow for 1.5 h and re-establishing the flow can be demonstrated.35 wt% AB/Si oil slurries are prone to separating when pumped using a peristaltic pump.Flow velocity changes at bends were observed to promote separation.Surface roughness/variations and plumbing transitions appear to promote phase separation.When AB particles start to separate, they form a porous plug allowing the oil to flow until the pores are completely blocked.When AB plugs develop, mechanical force is required to break it up (tubing was cut into sections to remove solidified AB and trapped Si oil).

### 7.5. Remaining Challenges and Opportunities

Partial success was observed in pumping fully hydrogenated silicone oil-based ammonia borane slurries (35 wt% AB) through Teflon tubing comprised of height changes and pipe bends. However, the reliable, efficient, and cost-effective transport of slurries with greater than 35 wt% solids loading is extremely unlikely considering the overall fuel cycle for automotive applications. Further complications are also expected if agglomeration, sintering, and density changes are realized with the dehydrogenated fuels. Given the technical challenges of slurry transport under ideal conditions, non-ideal conditions such as freezing temperatures (−40 °C), on- and off-boarding, and slurry stratification will only exacerbate the difficulties of slurry transport.

## 8. Batch Reactor Studies of Alane and Ammonia Borane Slurries

### 8.1. Objective

The objective of the batch reactor studies of alane and ammonia borane was to investigate potential solvent replacements for ammonia borane and alane slurry compositions that demonstrate improved dehydrogenation reaction kinetics, demonstrate facile transport on- and off-board, demonstrate long-term chemical compatibility, and demonstrate thermal stability within the expected dehydrogenation temperature range.

### 8.2. Technical Barriers and Challenges

Thermal stability of solvents;Chemical compatibility;Solvent effects on dehydrogenation kinetics and selectivity.

### 8.3. Approach

ATK alane and ammonia borane slurries were prepared using various solvents to investigate solvent effects on dehydrogenation rates. Alane slurries were prepared at 20 wt% using diethylene glycol dibutyl ether and Jalabo H350. Each alane slurry composition was also prepared with and without “catalysts”. The “catalyst” used was LiH and titanium (in the form of titanium t-butoxide). Ammonia borane slurries were prepared from silicone oil (AR−20) and IoLiLyte ionic liquid. The reactions were carried out in a stirred batch reactor with pressure monitoring capabilities.

### 8.4. Summary

In Figure 18 are the batch reactor results of “catalyzed” and “uncatalyzed” 20 wt% alane slurries with DEBE and Julabo H350 solvents. The choice of solvent had a greater impact on the alane dehydrogenation reaction rate than the addition of LiH/Ti. DEBE as a solvent was observed to enhance the dehydrogenation rate of alane over Julabo H350. For the case of DEBE-based slurries, small improvements were observed with the addition of LiH/Ti. However, the addition of LiH/Ti for the Julabo H350-based alane slurries did not demonstrate any marked improvement. The improvement may be a synergistic effect of the solvent–catalyst system.

Shown in Figure 19 are the batch reactor results of 20 wt% ammonia borane slurries with silicone oil and IoLiLyte ionic liquid. The IoLiLyte-based slurry showed some degree of enhancement with a lower dehydrogenation onset temperature. The ionic liquid-based slurry demonstrated a two-step reaction pathway and a lower overall maximum pressure achieved. Upon examination of the contents of the IoLiLyte slurry after completion, the final product was a solid yellow-colored foam: direct evidence that chemical compatibility is an issue with ammonia borane and IoLiLyte. The silicone oil-based slurry showed the typical three-step hydrogen release profile for ammonia borane. The sample did remain liquid-like, but the reaction products including polyborazylene settled to the bottom and required stirring to homogenize the composition.

### 8.5. Remaining Challenges and Opportunities

For non-automotive, niche applications using slurry-phase chemical hydrogen storage materials, the remaining challenges include optimizing the chemical compatibility, thermal stability, kinetics, and rheological characteristics of the liquid and solid-phase components. Optimizing the slurry compositions are expected to minimize reactor volumes, minimize reactor temperatures, and lastly, maximize reaction selectivity.

## 9. Fuel Cell Tolerance Testing: Diborane Impurity

### 9.1. Objective

The primary objective for this study was to determine the fuel cell tolerance level and fuel-cell degradation modes of diborane. Diborane is a known impurity that is generated during the dehydrogenation of ammonia borane.

### 9.2. Technical Barriers and Challenges

Fuel cell tolerance level;Fuel cell durability and operability.

### 9.3. Approach

Fuel cell durability measurements were performed with hydrogen feeds containing 40 ppm diborane. The three measurement techniques used to assess the effects of diborane on fuel cell operability were AC impedance spectroscopy (Figure 20A), VI or current-voltage curves (Figure 20B), and CV or cyclic voltammetry (Figure 20C).

### 9.4. Summary

Preliminary testing of a 40 ppm diborane–hydrogen mixture over a 20 h period indicated that the primary effect of diborane is on the charge-transfer resistance (CTR). The CTR increased with diborane exposure time. Diborane did not affect the active catalyst surface area; however, minor losses (~20 mV) were observed in the VI curve after a 20 h exposure of 40 ppm diborane. The fuel cell tolerance testing with 40 ppm diborane can be seen in Figure 20. Additional long-term testing is required to fully characterize the effects of diborane on fuel cell operability and durability.

### 9.5. Remaining Challenges and Opportunities

The challenge of all three classes of hydrogen storage materials (adsorbents, chemical, metal hydrides) is the validation testing of fuel cell tolerance with the expected impurities. This is especially true for metal hydrides, complex metal hydrides, and chemical hydrogen storage, and to a lesser extent, adsorbents. Identified fuel cell impurities will require purification components that add mass, volume, and complexity to the overall system. Additional impacts could include recycling valuable chemical constituents (e.g., boron).

## 10. Borazine Scrubbing

### 10.1. Objective

The objective of this study was to develop and demonstrate the feasibility of scrubbing borazine generated from ammonia borane dehydrogenation that prevents borazine and ammonia impurities from reaching the fuel cell thus impacting the long-term reliability and operability of the fuel cell.

### 10.2. Technical Barriers and Challenges

Borazine scrubbing capacity;Mass and volume of hydrogen purification components;Boron recovery and recycle;Borazine scrubber operation (replace or regenerate).

### 10.3. Approach

Our approach was to screen commercially available adsorbents for their affinity to adsorb borazine. Adsorbents of interest were zeolites and activated carbons. The adsorbents were tested under dynamic and static borazine environments in order to assess breakthrough and total adsorptive capacity. Borazine adsorption performance was investigated using as-received and chemically modified adsorbents. Adsorption performance was also investigated as function of temperature (25–150 °C).

### 10.4. Summary

Borazine adsorption was investigated on various zeolites and activated carbon adsorbents to quantify and compare the borazine adsorption capacities as a function of temperature and adsorbent conditioning (Figure 21). Activated carbon fiber ACN−210−15 was found to have the highest adsorption capacity (30–40 wt%). In comparison, the borazine adsorption capacities observed with zeolites w in the range of 3–12 wt% and 20–25 wt% for AX−21. As-received activated carbon fiber and as-received zeolites show higher borazine adsorption capacities as compared to the moisture-free substrate analogs (referred to as dry), suggesting that adsorbed water (referred to as wet) enhances borazine adsorption capacities. Borazine adsorption capacities were observed to decrease with increasing temperature. For ZSM−5, borazine adsorption capacity is not a strong function of surface acidity, but rather somewhat proportional to the specific surface area. In Y-zeolites, it is not the surface area, but a combination of high surface acidity and chemical form (H form) that appears to be favorable for borazine adsorption. Surface area is an important adsorbent parameter in identifying potential adsorbents. However, the surface functionalities are also important. Evidenced by comparing the borazine adsorption capacities for AX−21 and Zeolite Y, both adsorbents yielded roughly the same borazine adsorption capacity (0.015 g of borazine per gram of adsorbent). In comparison, the borazine adsorption capacity of ACN−210−15 was just under 0.03 g of borazine per gram of adsorbent. Higher borazine adsorption capacities observed in ACN−210−15 (as compared to AX−21) is thought to be related to the differing surface functionalities of the two activated carbons. Diffuse Reflectance Infrared Fourier Transform Spectroscopy (DRIFTS) studies of the adsorbate–adsorbent interactions indicate that the borazine is physisorbed on activated carbon adsorbents and chemisorbed on the zeolite adsorbents. Based on our research, ACN−210−15 activated carbon demonstrated the highest borazine adsorption capacity (both on mass and surface area basis) of all adsorbents tested. The fibrous form of ACN−210−15 makes this adsorbent ideally suited for environments such as vehicular travel where attrition resistance is necessary.

The borazine adsorbent ACN−210−20 was down-selected for further study. Figure 22 shows the breakthrough curves of borazine as a function of temperature. As expected, the borazine adsorption capacity decreases with increasing temperature.

Competitive adsorption studies of ammonia and borazine were carried to determine the order of adsorbent beds (Figure 23). The ammonia adsorbent bed was MnCl_2_/IRG−33 supplied by UTRC, and the borazine adsorbent bed was ACN−210−20 supplied by LANL. Ammonia was shown to have a low adsorption affinity for ACN−210−20 (Figure 23A). The total ammonia adsorption capacity of ACN−210−20 was around 1 wt%. In contrast, the total borazine adsorption capacity of ACN−210−20 was around 40 wt%. Shown in Figure 23B are the competitive adsorption results of ammonia and borazine with MnCl_2_/IRH−33. The total ammonia adsorption capacity of MnCl_2_/IRH−33 was around 6 wt%, while that for borazine was also around 6 wt%. Given the low adsorption affinity of ammonia on ACN−210−20, the order of the beds was determined to be the borazine adsorption bed (ACN−210−20) followed by the ammonia adsorption bed (MnCl_2_/IRH−33). Figure 23C shows the breakthrough curves of ammonia and borazine with a bed configuration of ACN−210−20 followed by MnCl_2_/IRH−33. The borazine breakthrough capacity was measured at 33 wt% (at time t_0_), and the ammonia breakthrough capacity was measured at 7 wt% (at time t_0_).

### 10.5. Remaining Challenges and Opportunities

Fuel cell impurities generated from metal hydrides and chemical hydrogen storage materials (to a lesser extent adsorbents) are problematic because of the added purification components required to produce fuel cell-grade hydrogen. Purification technologies are viable options in removing unwanted fuel cell impurities; however, the additional components impact the cost, mass, volume, and complexity of the overall system. The impacts are proportional to the identity and quantity of impurities generated. For example, the loss of nitrogen has a lesser impact on the overall fuel cycle than boron. Recycling or replenishing a high valued chemical constituent of the hydrogen storage material will impact regeneration costs and the overall efficiency of the fuel cycle. Small losses per vehicle quickly add up when considering the expected consumption rate of the entire vehicle fleet. The biggest challenge or opportunity is tailoring reaction chemistries or catalysis that maintain hydrogen reaction selectivities of one0(i.e., the only reaction occurring being the reaction producing hydrogen).

## 11. Ammonia Borane Ionic Liquid Compositions

### 11.1. Objective

The objective of this study using ionic liquids and ammonia borane was to investigate ionic liquids as potential solvents for ammonia borane fuel blends that have hydrogen gravimetric capacities in excess of 7.8 wt%.

### 11.2. Technical Barriers and Challenges

High solubility requirements of ammonia borane (≥−40 °C);High solubility requirements of spent fuel (≥−40 °C);Freezing point of solution (≥−40 °C);Chemical compatibility of ionic liquid with ammonia borane;Energy-efficient and cost-effective regeneration schemes of AB/IL compositions;Shelf-life of AB/IL compositions;Reaction kinetics and selectivity (impurities);Chemical compatibility;Thermal stability (>250 °C).

### 11.3. Approach

Ionic liquids were investigated to determine the feasibility of ammonia borane–ionic liquid compositions that can solubilize ammonia borane to produce hydrogen gravimetric capacities in excess of 6 wt%. The ionic liquids chosen were IoLiLyte, TbmpMS, DmimDmp, EmimAc, EmimDep, BmimCl, and EmimCl [28] The pertinent sought-after information included ammonia borane loading, ionic liquid viscosity, ionic liquid thermal stability, and AB/IL reactivity. We also looked at the dissolution of spent ammonia borane (i.e., polyborazylene) with the ionic liquids. The fuel composition must remain a liquid both pre- and post-reaction.

### 11.4. Summary

Of the ionic liquids surveyed, BmimCl and EmimCl were shown to yield the highest ammonia borane loadings of around 35 wt% (refer to Figure 24). However, no ionic liquid was observed to meet our target ammonia borane loading of 40 wt%. Binary mixtures of ionic liquids with IoLiLyte were shown to increase the ammonia borane loadings by as much as 50%. Eutectic solutions of ammonia borane with TbmpMs, EmimCl, and BmimCl were viscous solutions that could not be poured at room temperature. The same compositions upon dehydrogenation resulted in solid mixtures containing polyborazylene. The challenges with this approach include achieving and maintaining a high solubility loading at −40 °C (DOE minimum operating temperature) and also maintaining a liquid/fluid phase after dehydrogenation.

Viscosity measurements of the neat as-received and neat-dried ionic liquids were performed to ensure that the starting material did not exceed a viscosity of 1500 cP. The measured viscosities as a function of temperature of the neat ionic liquids are shown in Figure 25. IoLiLyte and EmimAc have the lowest viscosities (~75 cP at 30 °C), while EmimDep had the highest viscosity of around 190 cP at 30 °C. In all cases, the viscosities decreased with increasing temperature.

Thermogravimetric (TGA) measurements were performed on the ionic liquids to determine their thermal stability profiles as a function of temperature. Shown in Figure 26 are the TGA curves for EmimAc, EmimDep, DmimDmp, and TbmpMs. With the exception of TmpMs and IoLiLyte, all other ionic liquids were observed to decompose, thus making them unsuitable as dissolution media. TbmpMs was discontinued as a viable candidate owing to the fact that TbmpMs is a solid at room temperature, and because the spent fuel blend of AB/TbmpMs is a solid.

The TGA curve of the AB/IoLiLyte composition is shown in Figure 27. The salient features of Figure 27 are that no diborane was produced, only borazine and ammonia. Borazine is the primary impurity upon the release of the first equivalent of hydrogen, while ammonia is the primary impurity for hydrogen equivalents greater than one. After complete dehydrogenation of ammonia borane, the spent fuel blend of AB/IoLiLyte was a solid; consequently, the fuel composition was also discontinued as viable fuel ammonia borane fuel blend.

Various ionic liquids were investigated to determine their potential as ammonia borane solvents. Unfortunately, none of the ionic liquids surveyed resulted in ammonia borane fuel blends meeting all of the technical challenges and barriers with the primary shortcoming being the low solubility of ammonia borane. The most promising ionic liquids were the immidozolium halides and IoLilyte. Given the technical challenges and barriers encountered with a solvent (or slurry)-based ammonia borane composition, these approaches seem unlikely to meet all of the automotive requirements of a hydrogen storage material. The most promising hydrogen storage fuel is a neat liquid that remains a liquid after dehydrogenation.

### 11.5. Remaining Challenges and Opportunities

Solution-phase chemical hydrogen storage media require high solute mass fractions (~0.80) to achieve a net-usable gravimetric hydrogen capacity of around 7.8 wt%. The high solute mass fractions must be maintained at the low end of the operating temperature range (−40 °C). The low operating temperature is very demanding for all chemical hydrogen storage media containing a liquid component (slurries, solutions, or neat liquids). Solubility is critically important not only for the fully hydrogenated material but also of the partially and fully dehydrogenated material. In addition to solubility, chemical compatibility, thermal stability, boiling point, freezing point, and viscosity are also important material properties to be mindful of. Thermal breakdown of the solvent may cause cascading effects that include the following: fuel cell impurities requiring additional purification components, changes in solubility, boiling point depression, undesired volatile products, undesired side reactions with storage material, viscosity, the global life cycle performance, and the cost and efficiency of separating and replenishing the spent solvent.

## 12. Chemical Hydrogen Storage Material Properties

### 12.1. Objective

The objective of this study was to develop a set of fluid-phase chemical hydrogen storage material property guidelines for automotive applications meeting the 2017 DOE technical targets. The fluid-phase chemical hydrogen storage media considered in this study were neat liquids, solutions, and non-settling homogeneous slurries. The fluid-phase material property guidelines are expected to aid material researchers in their materials development and/or discovery efforts. Until now, the materials researchers relied on system level targets to guide their materials research; consequently, providing the materials research community with a viable set of chemical hydrogen storage materials properties fills a critical knowledge gap. Although the quantified set of material properties is not exhaustive, it is a necessary first step.

### 12.2. Technical Barriers and Challenges

Material gravimetric and volumetric capacities;Dehydrogenation kinetics;Regeneration efficiency;Impurities;System Efficiency;Enthalpy of dehydrogenation.

### 12.3. Approach

The methodology used for the determination of viable chemical hydrogen storage material properties is shown Figure 28. The ammonia borane system design developed by the Hydrogen Storage Engineering Center of Excellence was used as the baseline system design. The baseline system is presumed to contain the necessary components that will be common to all realizable fluid-phase chemical hydrogen storage media. Components of the ammonia borane system were identified as system independent (e.g., fuel cost), balance-of-plant (BOP)/material independent components (e.g., valves and pumps), and material dependent components (MDC, e.g., reactor). The term component is used loosely to not only represent physical components, but also quantities such as on-board efficiency and regeneration efficiency.

System-independent components are components invariant to the system design and vice-versa examples include fuel cost and regeneration efficiency. Material-independent components (referred to as balance-of-plant, BOP) include pumps, sensors, valves, tubing, etc. Because the BOP components were presumed to be material independent, the BOP components were grouped and treated as a constant with respect to mass, volume, durability, and operability. Material-dependent components (MDC) are the components whose mass and volume are reliant upon the material properties, kinetics, and thermodynamics of the chemical hydrogen storage media. Material-dependent components of interest are reactors, heat exchangers, volume displacement tanks, and hydrogen purification. To calculate viable material properties, the masses and volumes of the material d-pendent components were sized independent of the material. Given the a priori sizing of the MDC components and the BOP sizing, the minimum gravimetric and volumetric hydrogen capacities for slurries, solutions, and neat liquids were calculated that would meet the DOE 2017 gravimetric and volumetric system targets. A priori sizing of the MDC components permitted the calculation of material properties (e.g., kinetics and heat of reaction) that meet the mass and volume estimates of the material-dependent components (*vide infra*). The material properties guidelines described below are a summary of the critical findings presented in reference [2].

### 12.4. System Design

The generalized chemical hydrogen storage automotive system design used for this study is shown in Figure 29. The automotive system design captures the essential unit operations and ancillary components required for the storage, production, and delivery of fuel-cell grade hydrogen generated from fluid-phase chemical hydrogen storage media.

Two system design cases are presented in this study. Case 1 is the baseline case where the system design includes the envisaged system components necessary for automotive applications. The second case, Case 2, is an idealized case that has a lower system mass and system volume. The lower system mass and system volume for the idealized case stems from the fact that the hydrogen purification component was eliminated, and the dehydrogenation reactor was reduced by 50%. Eliminating the hydrogen purification component assumes the gas-phase reaction selectivity for hydrogen is one (i.e., hydrogen is the only gas-phase reaction product), and the vapor pressure of the liquid hydrogen storage media is negligible. The idealized case can be viewed as the lower bound with respect to system mass and system volume, correlating to a lower bound on the required hydrogen storage media capacity to meet the 2017 DOE technical targets. The system masses and volumes assumed for both the baseline and idealized systems can be seen in Table 5. Component masses and volumes assumed for this study will be detailed in later sections.

### 12.5. Summary

Realizable material properties related to chemical hydrogen storage for automotive applications meeting the 2017 DOE technical Table 6. The material properties were calculated assuming a generalized, material-independent, fluid-phase chemical hydrogen storage system design. The presented material properties were developed within the constraints of our system design, component sizing, assumptions, system operating conditions, and the DOE 2017 system targets. In addition, the ranges in material properties are not specific to a particular material and therefore can be applied to the general class of chemical hydrogen storage media. Viable material properties including gravimetric capacities, kinetics, heats of reaction, and impurity concentrations were calculated assuming neat liquids, solutions, and non-settling slurries. The material properties presented should not be taken as hard targets, but rather soft guidelines because of potential system trade-off scenarios. In fact, property values that fall within the presented material property values do not guarantee a commercializable media form. Likewise, material property values just outside the material property ranges do not imply that a material is not capable of meeting the system performance targets, but rather that the material will require further examination. Providing the materials research community with a viable set of chemical hydrogen storage material properties fills a critical knowledge gap in the search for novel chemical hydrogen storage media. The guidelines set forth, although not exhaustive, are an essential first step in identifying viable chemical hydrogen storage material properties, and most importantly, their implications on system mass, system volume, and system performance.

### 12.6. Remaining Challenges and Opportunities

The material property guidelines are those that were calculated to meet the DOE 2017 technical targets based on a material independent system design composed of the minimum number components required for a fully functioning automotive hydrogen storage system. The guidelines are not all inclusive; however, they do provide a necessary starting point for the development of chemical hydrogen storage materials. In short, the material meeting all of the chemical hydrogen storage properties simultaneously will ultimately be an outlier to our current thinking and approaches. The same can also be said of adsorbents and metal hydrides.

## 13. Conclusions

The DOE Hydrogen Storage Engineering Center of Excellence (HSECoE) was tasked with three objectives: (1) develop and validate chemical hydrogen storage system models, (2) determine viable chemical hydrogen storage material properties meeting the DOE system level targets, and (3) develop and demonstrate advanced engineering system components addressing challenges of the current state-of-the art chemical hydrogen storage media. As a whole, all three objectives were accomplished. Realistic system designs for both alane and ammonia borane were developed and used as a starting point for a more generalized material-independent system. The material-independent systems were the foundation for generating material properties that ultimately meet the U.S. Department of Energy’s (DOE) Technical System Targets: Onboard Hydrogen Storage for Light Duty Vehicles. The calculated chemical hydrogen storage material properties required to meet the DOE 2017 Technical System Targets are realizable from a thermodynamics, kinetics, and hydrogen capacity standpoint. The challenge is discovering the outlier of chemical hydrogen storage materials that possesses all of the material targets simultaneously.

System components were designed, built, and validated using the current state-of-the art chemical hydrogen surrogates alane and ammonia borane. These surrogates offered the opportunity to engineer solutions centered on the non-idealities of these two surrogates. The notable non-idealities that were successfully demonstrated include the mechanically mediated reactive transport of slurries, purification strategies for ammonia and borazine, and helical reactors designed to optimize conversion while preventing reactor slugging. Although chemical hydrogen storage materials currently suffer from the severe disadvantage with regeneration cost and efficiency, they do however offer high hydrogen gravimetric and hydrogen volumetric capacities, with hydrogen volumetric capacities expected to surpass the DOE Technical System Targets. Neat liquid-phase chemical hydrogen storage media are, in all likelihood, the only commercializable media form for automotive applications.

## Figures and Tables

**Figure 1 molecules-26-01722-f001:**
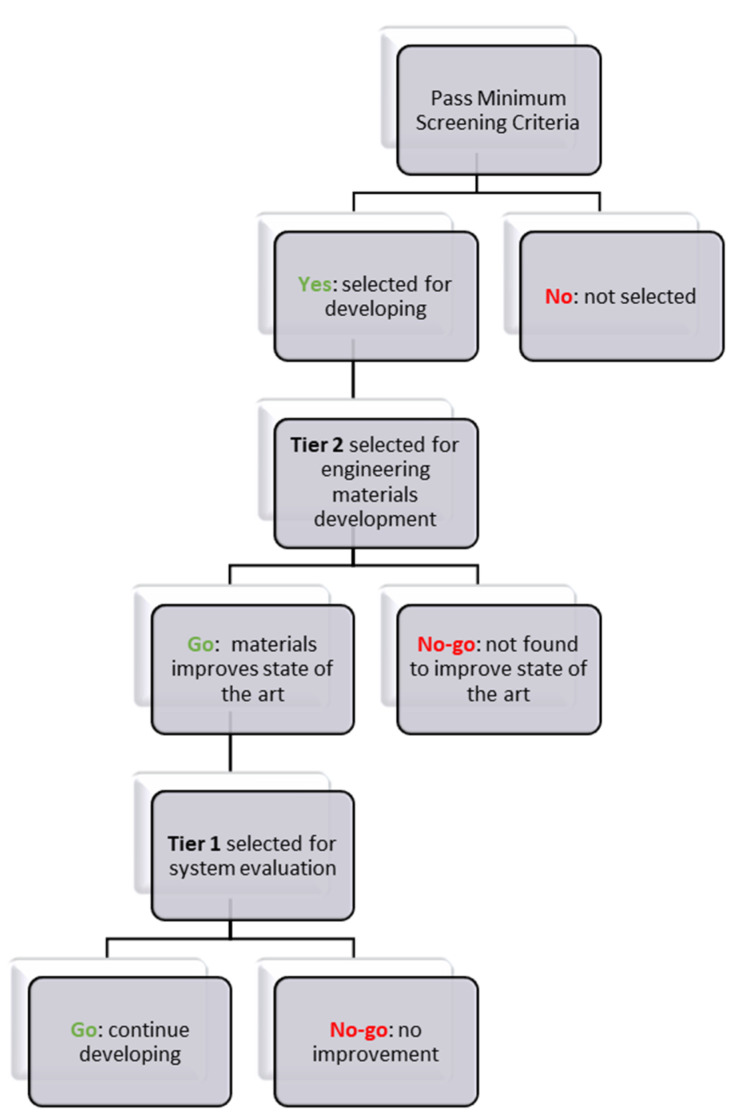
Materials selection procedure developed by the Hydrogen Storage Engineering Center of Excellence (HSECoE) to select candidates for engineering materials development and further system evaluation.

**Figure 2 molecules-26-01722-f002:**
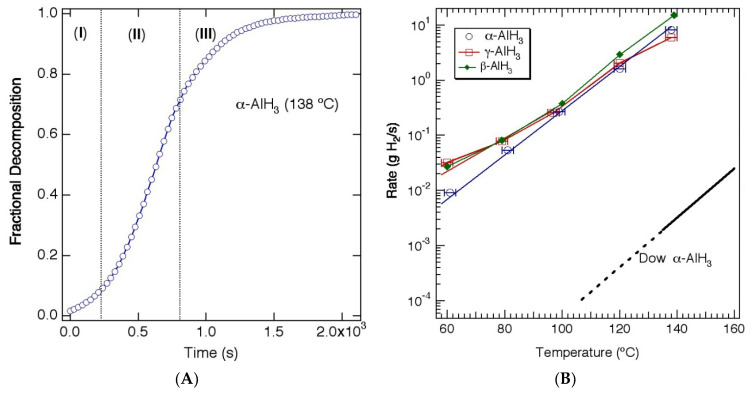
(**A**) Isothermal decomposition of α-AlH_3_ (Dow) at 138 °C showing (I) induction period, (II) acceleratory period, and (III) decay period. (**B**) Rate of H_2_ release as a function of temperature during thermal decomposition of α, β, and γ-AlH_3_ and α-AlH_3_ prepared by the Dow Chemical Company. Reproduced from [10]. (Adapted with permission from Elesevier).

**Figure 3 molecules-26-01722-f003:**
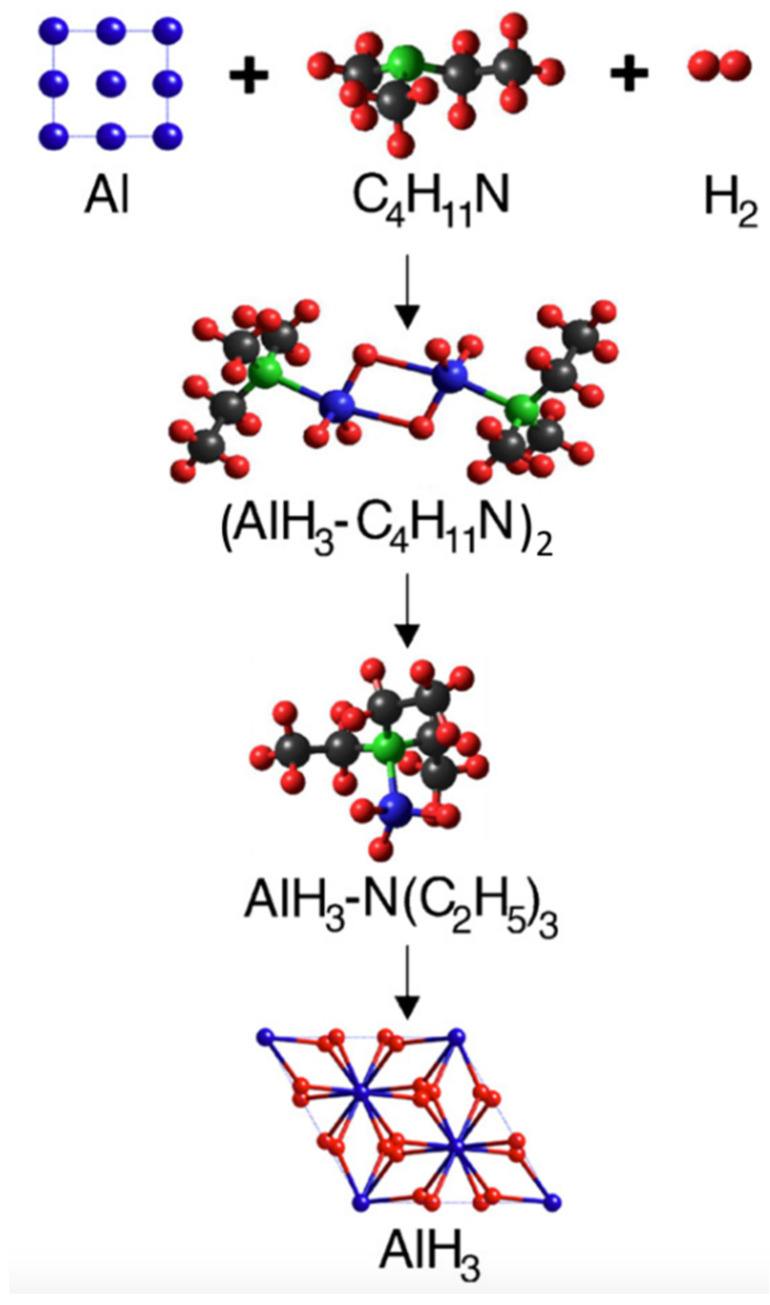
Example of a regeneration pathway for AlH_3_ showing hydrogenation of Al and dimethylethylamine (DMEA) to form DMEA-AlH_3_, followed by incorporation of triethylamine (TEA) and transamination to form TEA-AlH_3_, followed by separation to recover AlH_3_. Reproduced from [6]. (Adapted with permission from Elesevier).

**Figure 4 molecules-26-01722-f004:**
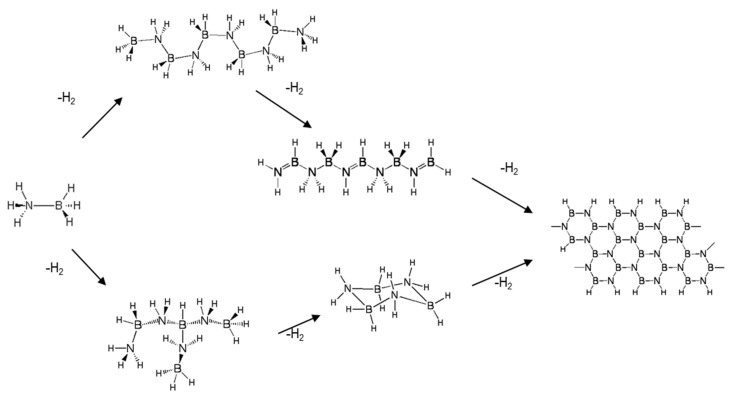
Reaction pathways and products observed during the thermal decomposition of solid-phase ammonia borane [3].

**Figure 5 molecules-26-01722-f005:**
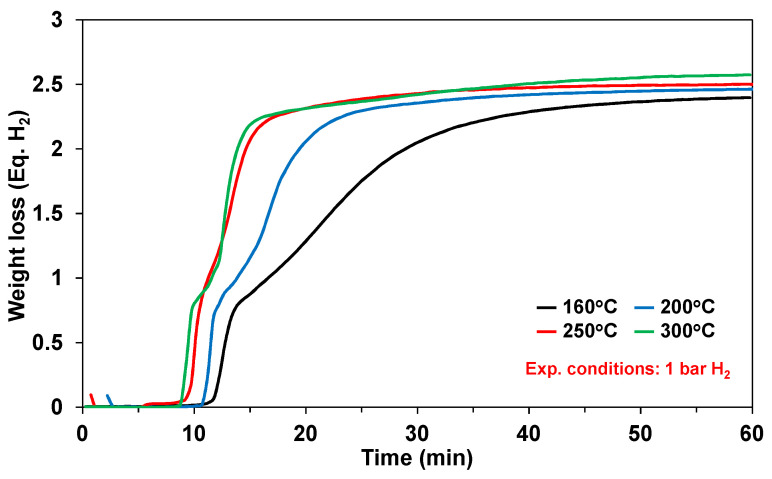
Measurements of hydrogen capacity and bulk kinetics of solid ammonia borane (AB) at 160 °C (black), 200 °C (blue), 250 °C (red), and 300 °C (green). Reproduced from [7].

**Figure 6 molecules-26-01722-f006:**
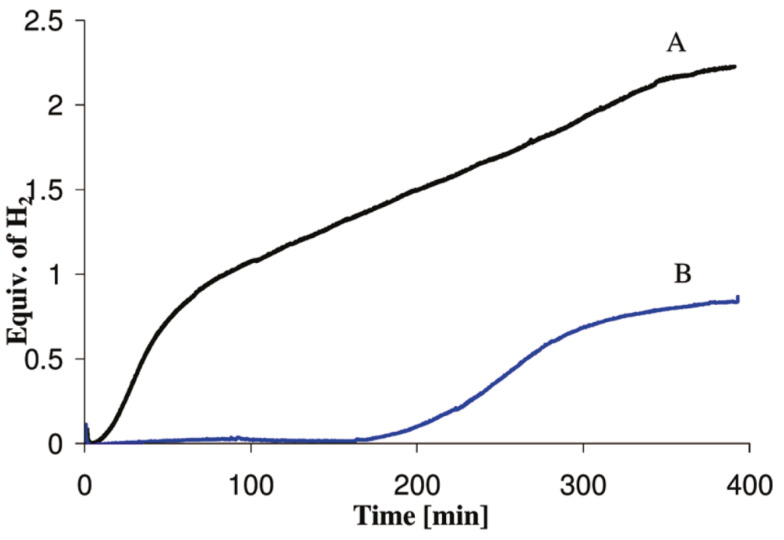
Hydrogen release measurements via gas buret at 85 °C for a 50:50 mixture of AB (150 mg):BmimCl (curve A) and 150 mg of neat solid-phase AB (curve B); data reproduced from [22].

**Figure 7 molecules-26-01722-f007:**
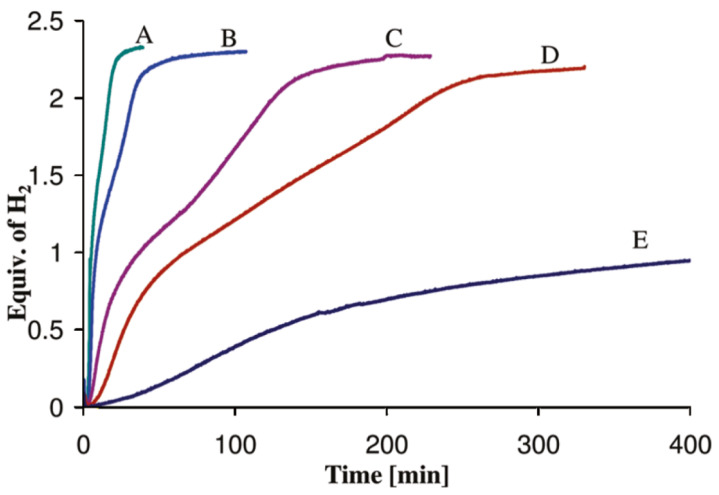
Hydrogen release measurements via gas burret for a 50:50 mixture of AB (150 mg):BmimCl mixture as a function of isothermal temperatures: (**A**) 110 °C, (**B**) 105 °C, (**C**) 95 °C, (**D**) 85 °C, and (**E**) 75 °C; data reproduced from [22].

**Figure 8 molecules-26-01722-f008:**
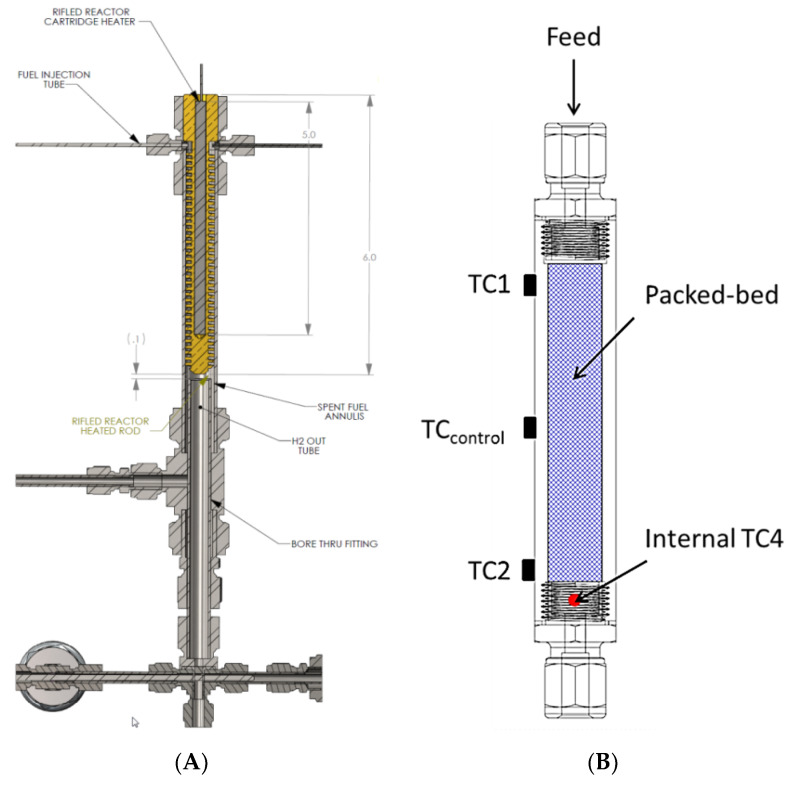
Helical (**A**) and packed-bed (**B**) reactors used for liquid-phase methoxy propyl amine borane hydrogen storage media. Note: TC denotes thermocouple.

**Figure 9 molecules-26-01722-f009:**
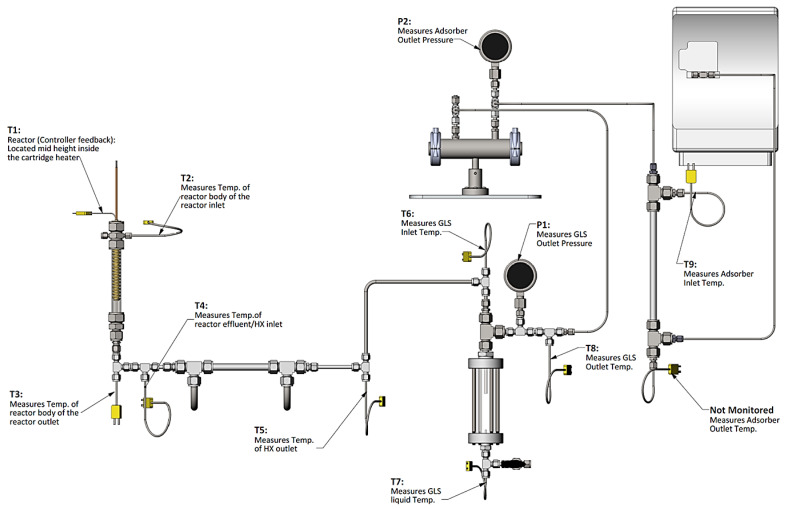
Reactor test bed apparatus equipped with reactor, heat exchanger, gas–liquid separator (GLS), impurity adsorbent bed, and gas-phase analytical components. Note: T denotes thermocouple, and P denotes pressure gauge.

**Figure 10 molecules-26-01722-f010:**
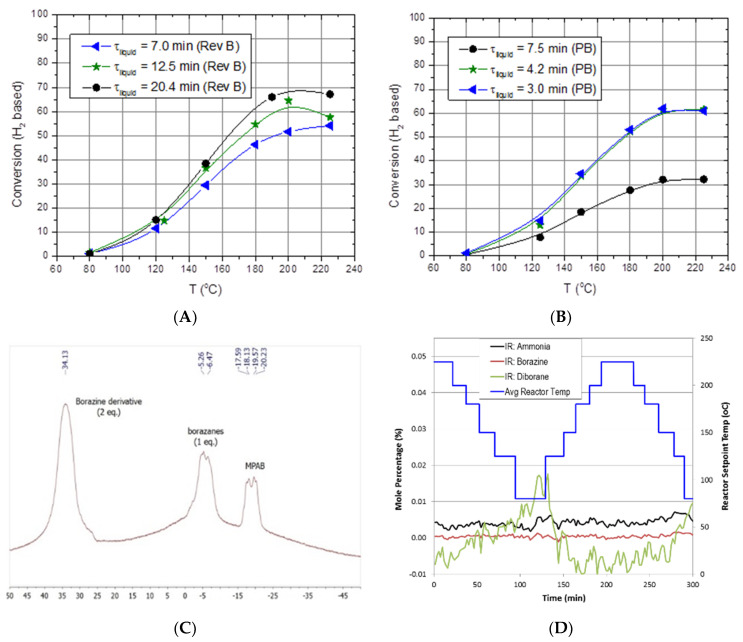
Representative results of methoxypropyl amine borane (MPAB). (**A**) Conversion as a function of reactor temperature and space–time for helicial reactor, (**B**) conversion as a function of reactor temperature and space–time for SiC packed bed (PB) reactor, (**C**) NMR analysis of liquid-phase products, (**D**) in situ gas-phase IR analysis of impurities generated from methoxypropyl amine borane (MPAB) dehydrogenation.

**Figure 11 molecules-26-01722-f011:**
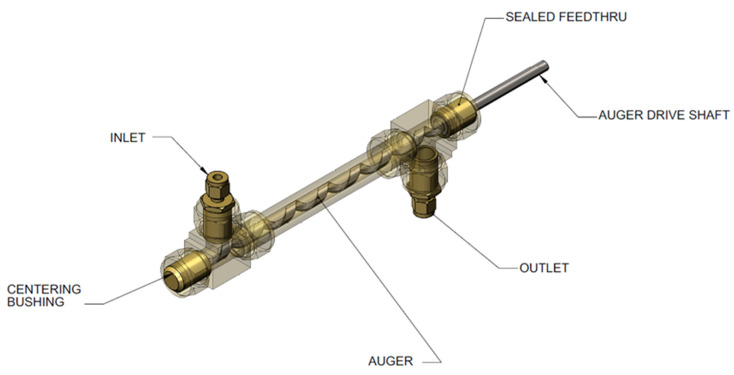
Auger reactor used for alane and ammonia borane (AB) slurry feeds.

**Figure 12 molecules-26-01722-f012:**
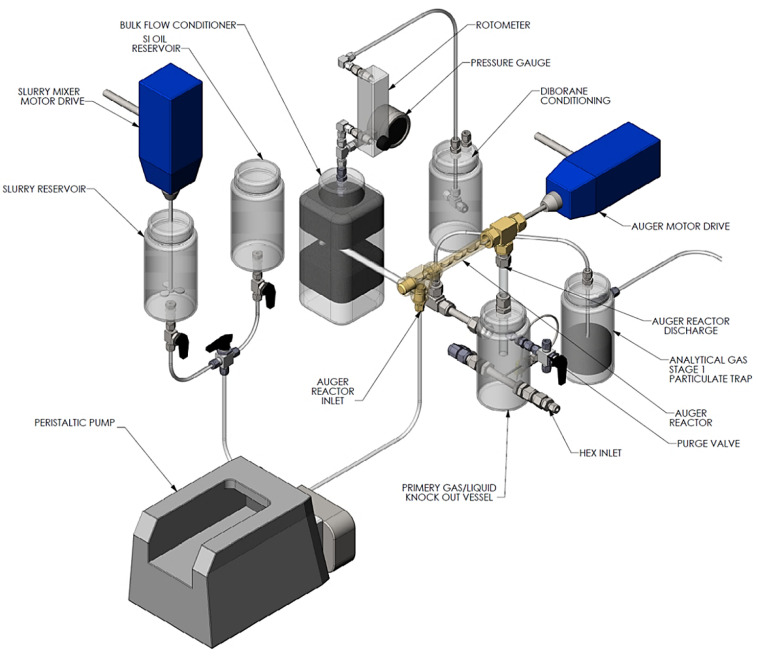
Schematic of slurry feed system, auger reactor, heat exchanger, and gas–liquid separators for slurry-based hydrogen storage media (gas-phase analytics not shown).

**Figure 13 molecules-26-01722-f013:**
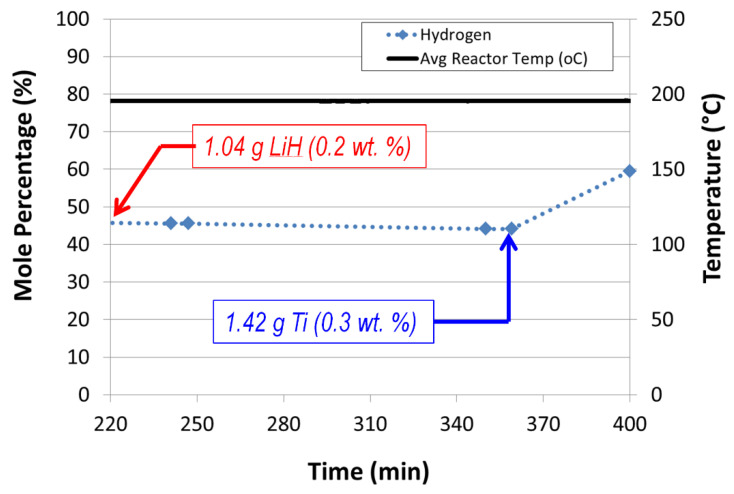
Effects of sequential addition of LiH and Ti t-butoxide on alane dehydrogenation in our auger reactor (60 wt% ATK alane slurry in mechanical pump fluid, auger speed = 12 rpm, feed flow rate = 1.02 mL/min).

**Figure 14 molecules-26-01722-f014:**
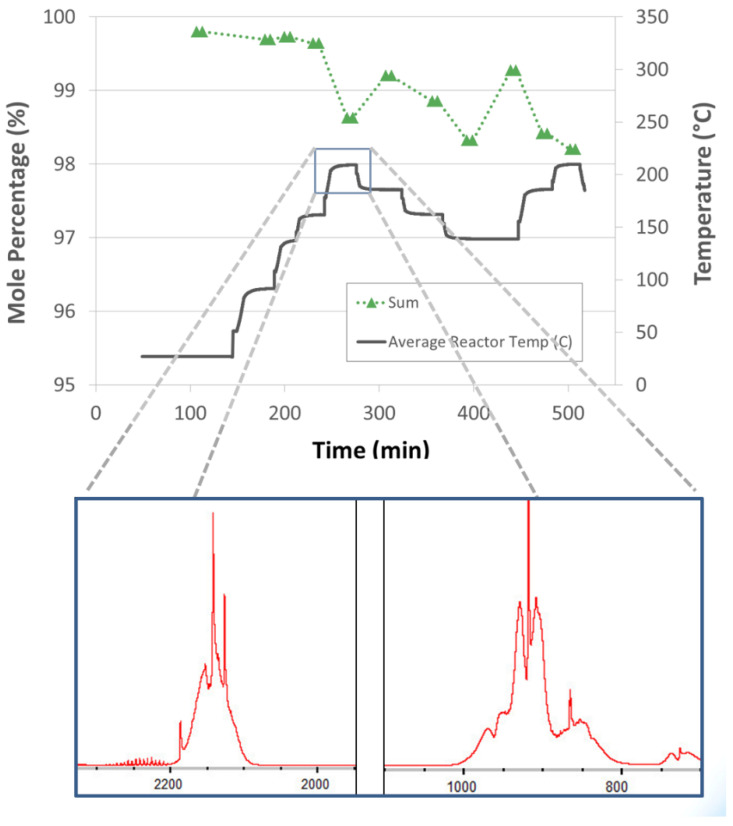
IR spectra of Si-H transitions observed with silicone oil-based alane slurries for reactor temperatures around 200 °C indicating an alane/aluminum side reaction with the silicone oil carrier (abscissa units are cm^−1^).

**Figure 15 molecules-26-01722-f015:**
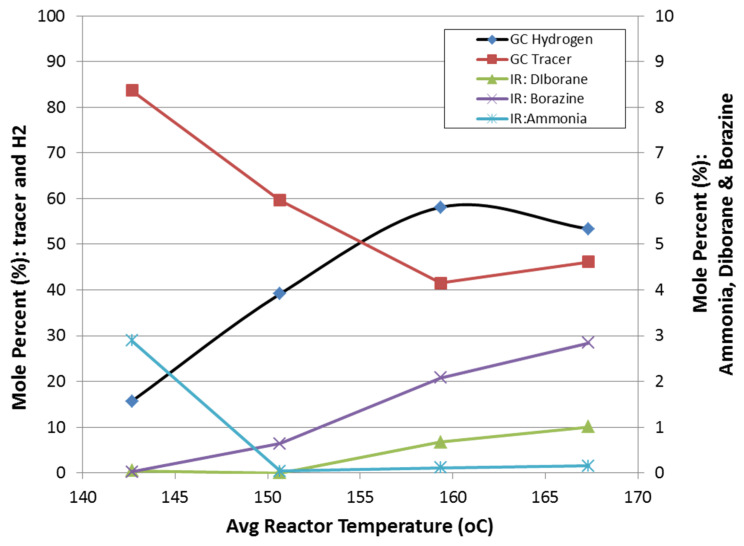
Reaction selectivity of ammonia borane dehydrogenation as a function of reactor temperature.

**Figure 16 molecules-26-01722-f016:**
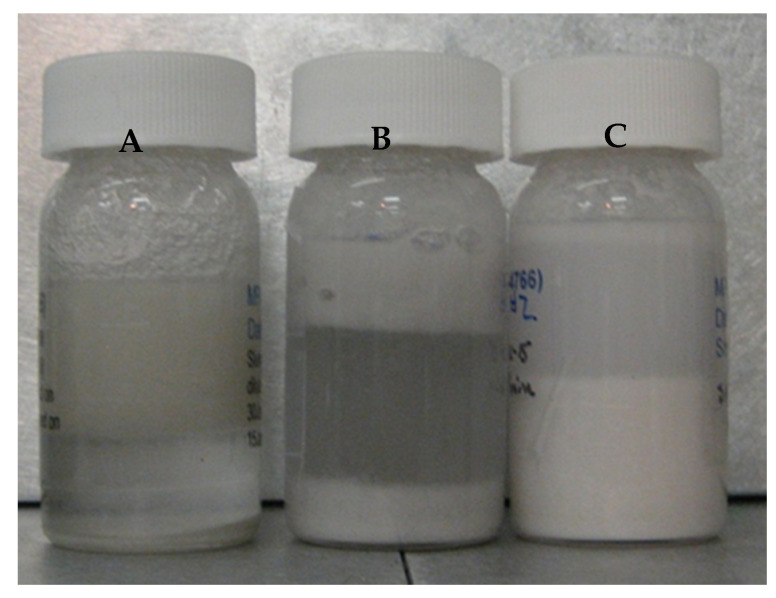
Ammonia borane slurry stratification based on the extent of ammonia borane conversion of as-prepared silicone oil-based ammonia borane slurry (**A**), partially reacted AB-Si oil slurry (**B**), and (**C**) >90% ammonia borane conversion.

**Figure 17 molecules-26-01722-f017:**
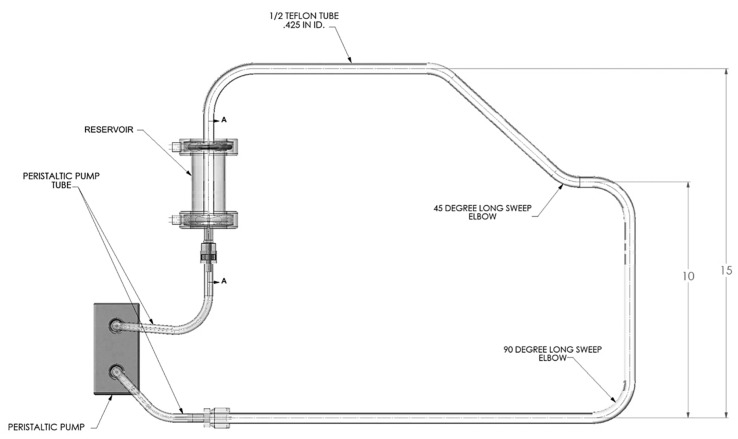
Ammonia borane slurry transport testing apparatus. Length scale is in inches.

**Figure 18 molecules-26-01722-f018:**
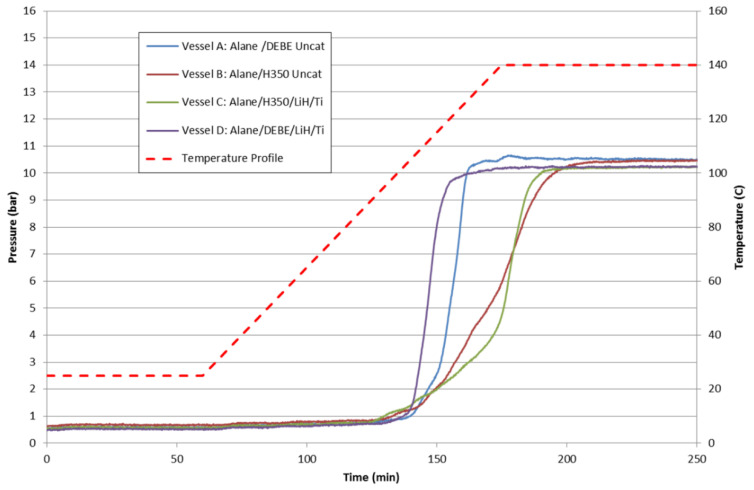
Batch reactor results of 20 wt% alane slurries for alane/DEBE uncatalyzed (blue), alane/DEBE catalyzed with LiH/Ti (purple), alane/Julabo H350 uncatalyzed (red), and alane/Julabo H350 catalyzed with LiH/Ti (green).

**Figure 19 molecules-26-01722-f019:**
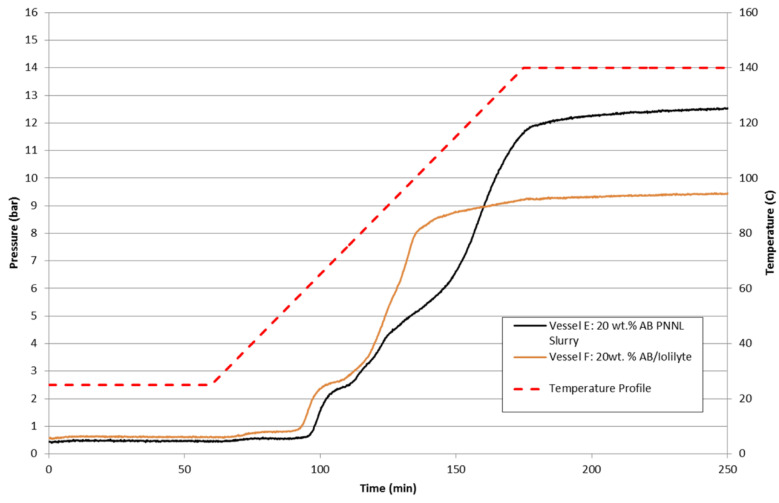
Batch reactor results of 20 wt% ammonia borane slurries in AR−20 silicone oil (black) and ionic liquid IoLiLyte (orange).

**Figure 20 molecules-26-01722-f020:**
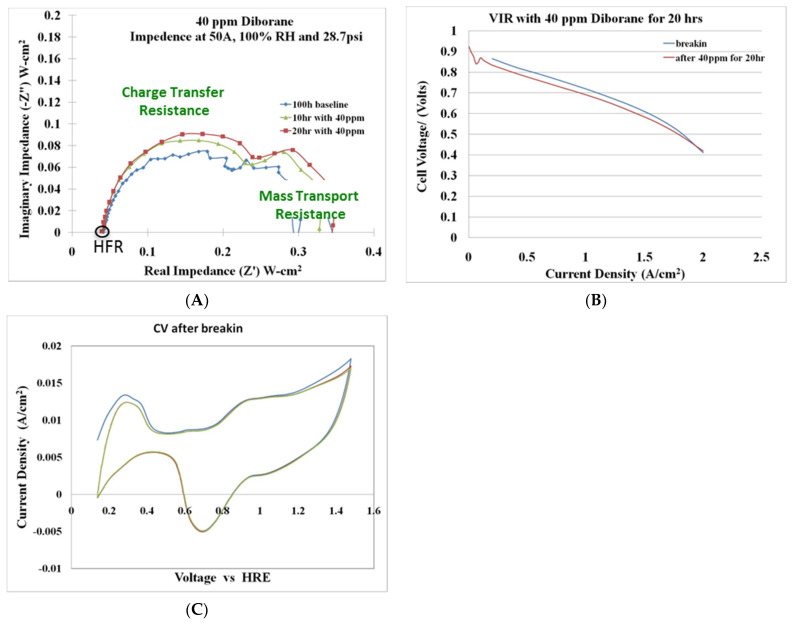
Fuel cell tolerance testing with 40 ppm diborane (**A**) AC impedance spectroscopy, (**B**) current-voltage-resistance (VIR) curves, and (**C**) cyclic voltammograms for a 20 h exposure test.

**Figure 21 molecules-26-01722-f021:**
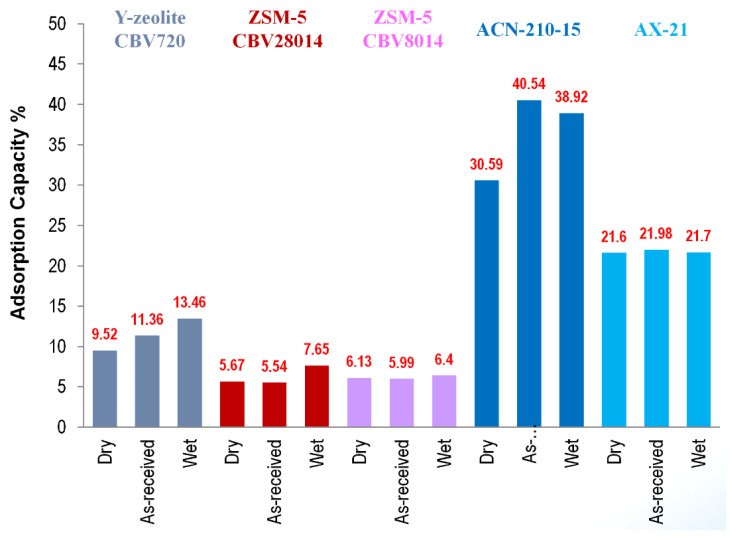
Borazine adsorption capacities for various adsorbents and adsorbent treatments (adsorption capacity measured at room temperature).

**Figure 22 molecules-26-01722-f022:**
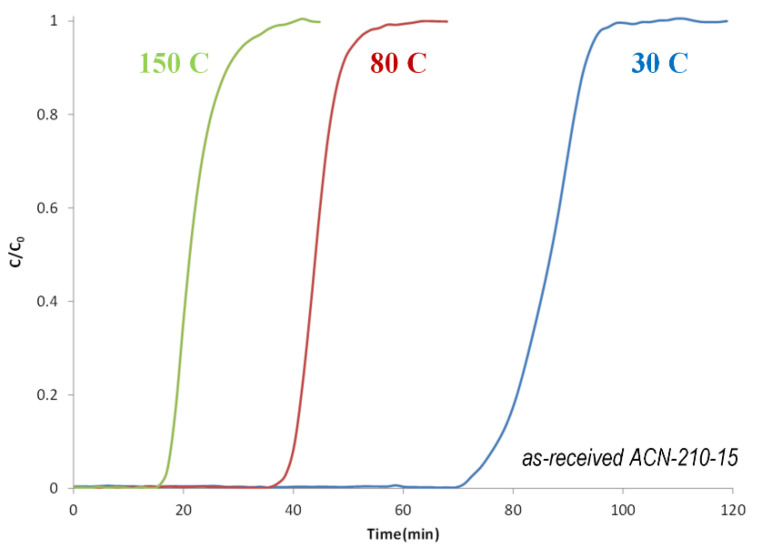
Borazine breakthrough curves as a function of temperature for as-received ACN−210−20.

**Figure 23 molecules-26-01722-f023:**
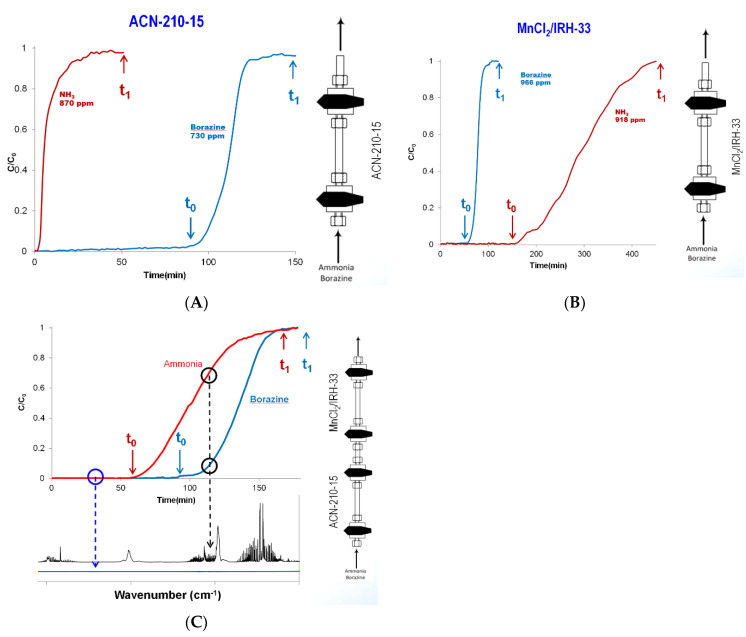
Competitive room temperature adsorption studies of ammonia and borazine with (**A**) ACN−210−20, (**B**) MnCl_2_/IRH−33, and (**C**) both ACN−210−20 and MnCl_2_/IRH−33; (t_0_ = time to initial breakthrough, t_1_ = time to complete bed saturation or total adsorption capacity).

**Figure 24 molecules-26-01722-f024:**
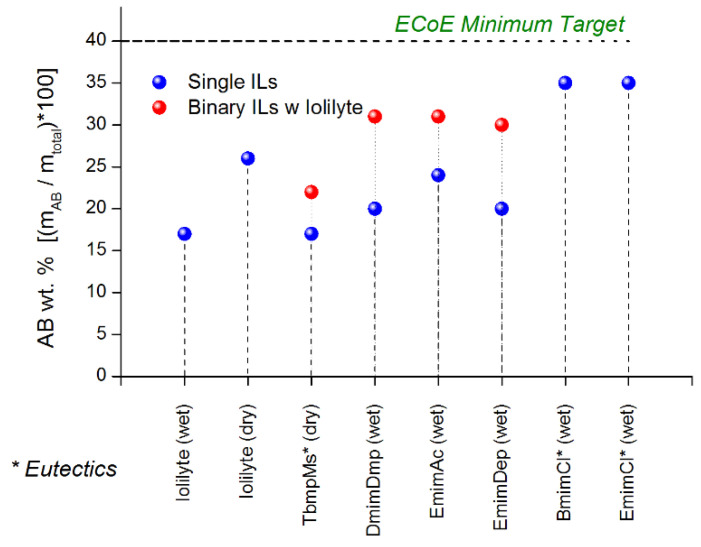
Maximum ammonia borane loadings for various ionic liquids and binary ionic liquids. Reproduced from [28].

**Figure 25 molecules-26-01722-f025:**
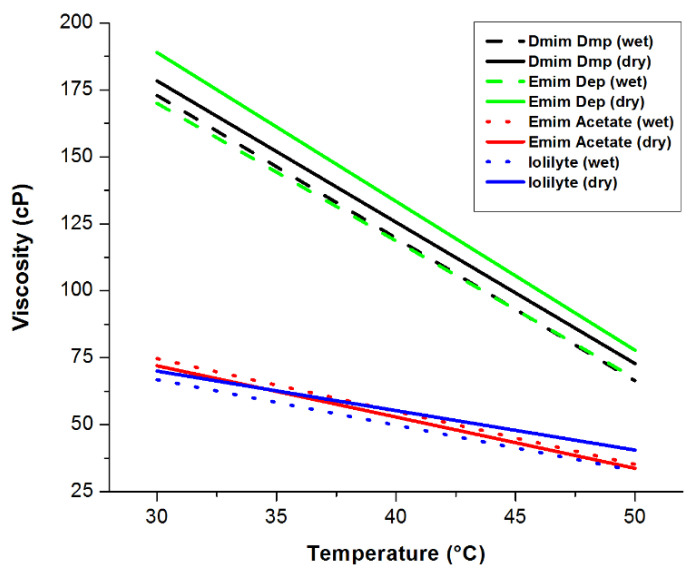
Viscosity measurements as a function of temperature of neat ionic liquids. Reproduced from [28].

**Figure 26 molecules-26-01722-f026:**
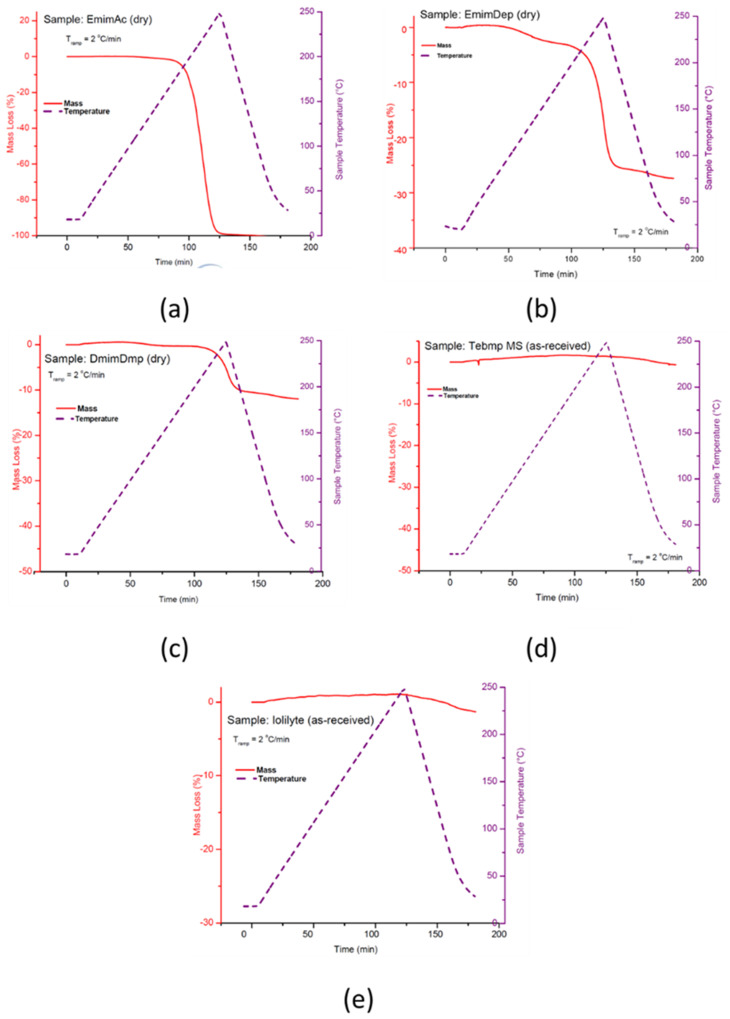
TG decompositions curves of neat ionic liquids for (**a**) EmimAc, (**b**) EmimDep, (**c**) DmimDmp, (**d**) TbmpMs, and (**e**) IoLiLyte. Reproduced from [28].

**Figure 27 molecules-26-01722-f027:**
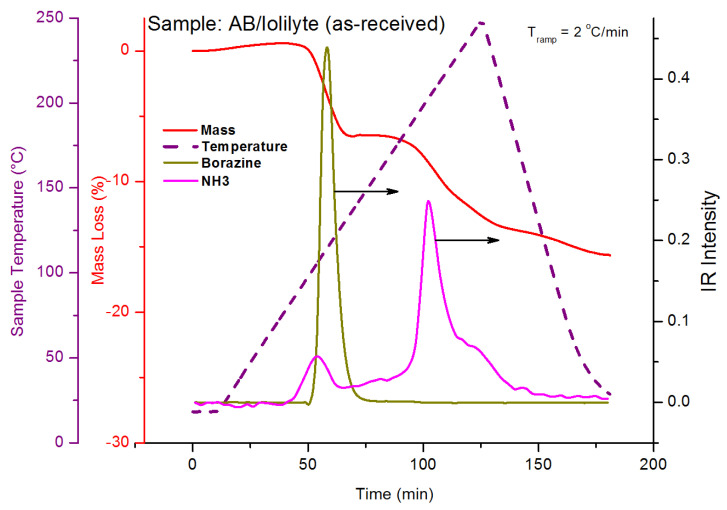
TG-IR curves for decomposition of ammonia borane–IoLiLyte composition. Reproduced from [28].

**Figure 28 molecules-26-01722-f028:**
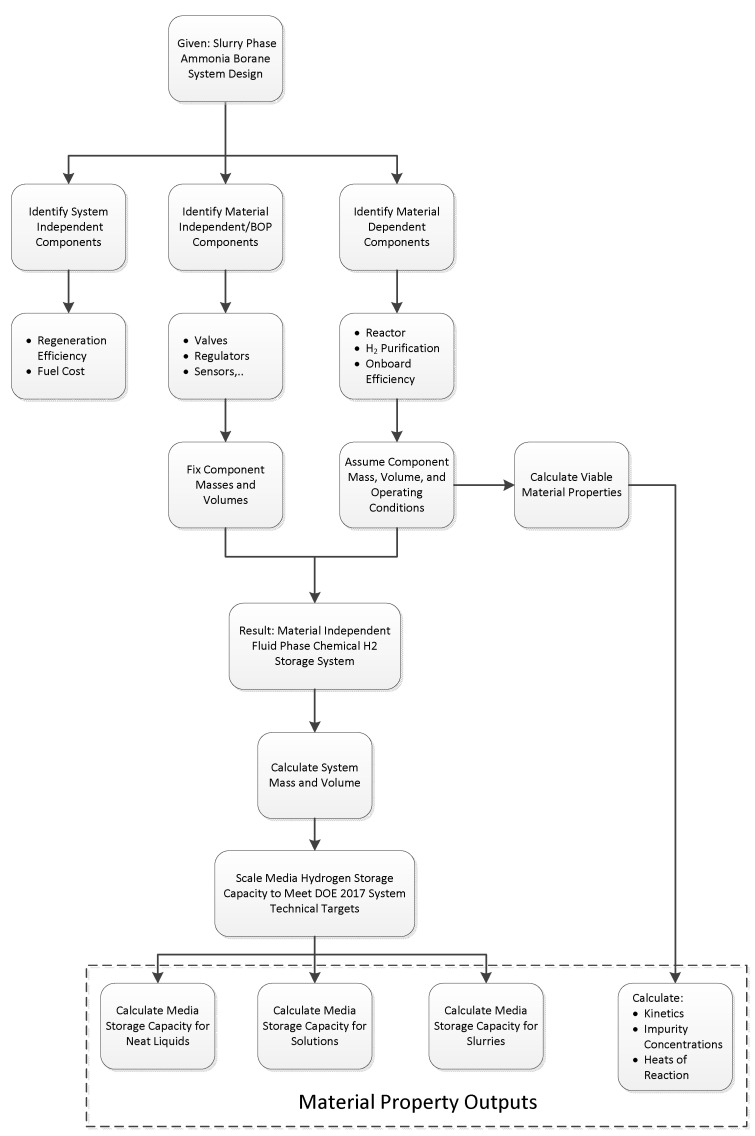
Modeling methodology flow chart.

**Figure 29 molecules-26-01722-f029:**
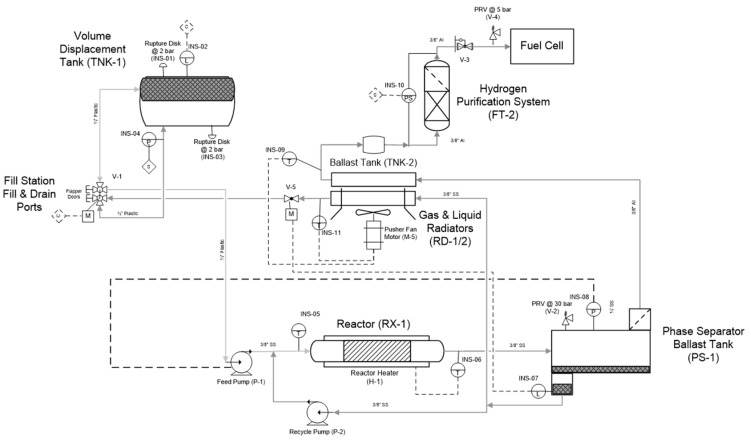
Automotive system design for off-board regenerable chemical hydrogen storage media [2].

**Table 1 molecules-26-01722-t001:** U.S. Department of Energy (DOE) targets for on-board hydrogen storage systems for light-duty vehicles [1].

Scheme 2010	Units	2010	2017	Ultimate
System Gravimetric Capacity: usable, specific-energy from H_2_ (net useful energy/max system mass)	kWh/kg (kg H_2_/kg system)	1.5 (0.045)	1.8 (0.055)	2.5 (0.075)
System Volumetric Capacity: usable, energy density from H_2_ (net useful energy/max system volume)	kWh/L (kg H_2_/L system)	0.9 (0.028)	1.3 (0.040)	2.3 (0.070)
Storage System Cost	$/kWh net ($/kg H_2_)	TBD (TBD)	TBD (TBD)	TBD (TBD)
• Fuel cost	$/gge at pump	3–7	2–4	2–4
Durability/Operability				
• Operating ambient temperature	°C	−30/50 (sun)	−40/60 (sun)	−40/60 (sun)
• Min/max delivery temperature	°C	−40/85	−40/85	−40/85
• Operational cycle life (1/4 tank to full)	Cycles	1000	1500	1500
• Min delivery pressure from storage system; FC = fuel cell, ICE = internal combustion engine	bar (abs)	5 FC/100 ICE	5 FC/100 ICE	5 FC/100 ICE
• Max delivery pressure from storage system	bar (abs)	12 FC/100 ICE	12 FC/100 ICE	12 FC/100 ICE
• Onboard Efficiency	%	90	90	90
• “Well” to Powerplant Efficiency	%	60	60	60
Charging/Discharging Rates:				
• System fill time (5 kg)	min (kg H_2_/min)	4.2 (1.2)	3.3 (1.5)	2.5 (2.0)
• Minimum full flow rate	(g/s)/kW	0.02	0.02	0.02
• Start time to full flow (20 C)	s	5	5	5
• Start time to full flow (−20 C)	s	15	15	15
• Transient response: • 10% −90% and 90%−0%	s	0.75	0.75	0.75
Fuel Purity (H_2_ from storage)	% H_2_	SAE J2719 and ISO/PDTS 14687–2 (99.97% dry basis)		
Environmental Health & Safety				
• Permeation and leakage	Scch	Meets or exceeds applicable standards		
• Toxicity	-			
• Safety	-			
• Loss of useable H_2_	(g/h) kg H_2_ stored	0.1	0.05	0.05
Useful constants: 0.2778 kWh/MJ; 33.3 kWh/kg H_2_; 1 kg H_2_ ≈ 1 gal gasoline equivalent (GGE)				

**Table 2 molecules-26-01722-t002:** Minimum screening criteria for each materials group.

Materials Group	Minimum Screening Criteria
Metal Hydrides	Capacity: >9 wt% materials capacity to be able to meet the DOE 2015 system target Absorption: Room temperature (RT) to 250 °C at 1–700 bar H_2_ pressure, rate >20 g/s (storing 5 kg accessible H_2_) Desorption: 80 °C to 250 °C at 1–3 bar H_2_ pressure, rate > 20 g/s (5 kg net usable H_2_) Enthalpy: <50 kJ/mol Crystal density: >1 g/cm^3^
Chemical Hydrogen Storage Materials	Capacity: >9 wt% materials capacity to be able to meet the DOE 2015 system target Desorption: RT to 150 °C, rate > 30 g/s (storing 5 kg accessible H_2_) Enthalpy: <20 kJ/mol Crystal density: >1 g/cm^3^ Availability: quantitative cost and time, i.e., <USD 10,000/kg in 30 day delivery
Adsorbent Materials	Capacity: Max Gibbs excess capacities >5 wt% and 30 g/L with storage T′s from 77 K to RT and <50 bar at 77 K Desorption: measured between 80 K and RT, must meet DOE 2010/2015 target of 0.02 g/s/kW Hydrogen uptake: measured between 80 K and RT, <30 bar, must be 30 g H_2_/s Specific Surface Area: Prefer 3000 m^2^/g, pore sizes 0.7 to 1.5 nm, pore volume 1.2 cm^3^/g Bulk density: >0.7 g/cm^3^

**Table 3 molecules-26-01722-t003:** Materials candidates grouped into Tier 1, Tier 2, and not further considered materials.

Materials Group	Tier 1 Developed Materials	Tier 2 Developing Materials	Not Further Considered Materials
Adsorbents	AX−21 MOF 5	Pt/AC-IRMOF 8	MOF 177
Chemical Hydrides	NH_3_BH_3_ (s) AlH_3_	NH_3_BH_3_ (l) LiAlH_4_	
Metal Hydrides	NaAlH_4_ 2LiNH_2_ + MgH_2_	Mg(NH_2_)_2_ + MgH_2_ + 2LiH TiCr(Mn)H_2_	MgH_2_ Mg_2_NiH_4_

**Table 4 molecules-26-01722-t004:** Evaluation of materials operating requirements for fluid-phase ammonia borane compositions.

Scheme	Minimum Criteria	Quantified Properties for Liquid AB	Quantified Properties for AB Slurry
Hydrogen Content of the ChemicalHydrogen Storage Material (includes slurrying agents and additives)	≥6 wt%	4.5–6.0 wt% at 2.35 H_2_ Equivalents	~7.2 wt% obtained (45 wt% loading)
Phase change	No phase change that would result in material not being transportable	Temperature and Space-time dependent as well as composition dependent	No phase change observed under current conditions
Flow ability before and after H-release at −20 °C to 40 °C	<1500 cP	Typical ranges Fresh: 100–250 cP Spent: 100–(>1500 cP)	~600 cP for 45% AB fresh and spent slurry
Thermal and chemical stability at −20 °C to 40 °C	Shelf life >6 weeks	1.8% (0.04 H_2_ Eq) reacted in two weeks @ 30 °C	Fresh AB slurry stable at r.t. for >4 months
On-Board Efficiency based on HSECoE drive cycles and up to 8 start-ups per day	>90%	>90% (modeling outputs)	>90%; Modeling shows feasibility
Safety	Safety issues comparable to gasoline or Safety issues can be mitigated reasonably	Analogous to solid AB	Analogous to solid AB. Flammability test was performed
Kinetics	Similar to neat AB	Better or equivalent to solid AB	Better or equivalent to neat AB since there is no induction period
Settling/flocculation	Minimum mitigation/stirring to recover sample acceptable	Dissolved media-no issues; Slurry feed will flocculate but spent fuel is a solution	Fresh AB (40%) good for >4 months. Spent AB settles within hours; stirring to recover acceptable
Heat of reaction	Similar to neat AB	Comparable to solid AB	Similar to AB
Impurities	Can be mitigated reasonably	Strategies exist to purify H_2_. The critical aspect is the adsorbent sizing Maximum Impurities Borazine: 2–4 mol% Ammonia: 0.25–0.5 mol%	Strategies exist to purify H_2_
Scale-up to larger quantities	≥1 L can be produced	Yes	Yes, achievable

**Table 5 molecules-26-01722-t005:** Baseline and idealized system component masses and volumes for fluid-phase chemical hydrogen storage media used. Reproduced from [2].

Component	Baseline Case		Idealized Case	
	Mass (kg)	Volume (L)	Mass (kg)	Volume (L)
Media + Tank	≤65.5	≤98.9	≤71.5	≤104.9
Reactor ^a^	5	4	2.5	2
H_2_ Purification ^a^	3.2	4	0	0
Heat Exchangers ^a^	3.7	9.2	3.7	9.2
Ballast Tank ^a^	2.6	15	2.6	15
BOP ^b^	21.8	8.9	21.8	8.9

^a^ component masses or volumes were sized independent of material to maintain a material independent system; ^b^ BOP mass and volume were fixed for both system cases.

**Table 6 molecules-26-01722-t006:** Summary of chemical hydrogen material properties required to meet the 2017 DOE Targets for Onboard Hydrogen Storage Systems for Light Duty Vehicles [2].

Parameter	Symbol	Units	Range *	Assumptions
Minimum Material capacity (liquids)	γ_mat_	g _H2_/g _material_	~0.078 (0.085) ^†^	• System mass (excludes media) = 30.6 kg (36.3 kg) • 5.6 kg of net-usable hydrogen • Liquid media (neat) • Media density = 1.0 g/mL
Minimum Material capacity (solutions)	γ_mat_	g _H2_/g _material_	~0.098 (0.106) ^†^	• System mass (excludes media) = 30.6 kg (36.3 kg) • Solute mass fraction = 0.80 • Solution density = 1.0 g/mL
Minimum Material capacity (slurries)	γ_mat_	g _H2_/g _material_	~0.112 (0.121) ^†^	• System mass (excludes media) = 30.6 kg (36.3 kg) • Non-settling homogeneous slurry • Slurry mass fraction = 0.70 • Slurry volume fraction = 0.5 • Slurry density = 1.0 g/mL
Kinetics: Activation Energy	E_a_	kJ/mol	117–150	• Reactor volume ≤ 4 L • Shelf life ≥ 60 days • Reaction order, n = 0.13 (100 °C), 0.5 (125 °C), 1 (175 °C), 2 (300 °C)
Kinetics: Preexponential Factor	A		4 × 10^9^–1 × 10^16^	
Endothermic Heat of Reaction	ΔH_rxn_	kJ/mol H_2_	≤+17 (15) ^†^	• On-board Efficiency = 90% • Number of cold startups = 4 • ΔT = 150 °C with no heat recovery • Neat liquid (Cp = 1.6 J/g K) • Reactor mass = 2.5 kg SS (5.0 kg SS)
Exothermic Heat of Reaction	ΔH_rxn_	kJ/mol H_2_	≤−27	• Maximum Temperature = 250 °C • Recycle ratio = 50%
Maximum Reactor Outlet Temperature	T_outlet_	°C	250	• Liquid radiator = 2.08 kg • Gas radiator = 0.3 kg • Ballast tank = 2.6 kg
Impurities Concentration	y_i_	ppm	No a priori estimates can be quantified	• Adsorbent mass ≤3.2 kg
Media H_2_ Density	(γ_mat_)(ф_m_)(ρ_mat_)	kg H_2_/L	≥0.07	• HD polyethylene tank ≤ 6.2 kg
Regen Efficiency	η_regen_	%	≥66.6%	• On-board Efficiency = 90% • WTPP efficiency = 60%

* (a) parameter values are based on a specific system design and component performance with fixed masses and volumes (b) values outside these ranges do not imply that a material is not capable of meeting the system performance targets; (c) the material property ranges are subject to change as new or alternate technologies and/or new system designs are developed; (d) the minimum material capacities are subject to change as the density of the composition changes due to reductions in the mass and volume of the storage tank or reductions in system mass are realized; ^†^ values outside of parentheses are the values that correlate to the idealized system design (i.e., 30.6 kg) and the values in parentheses are those that correlate to the baseline system design (36.3 kg).

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
