# Peer review of "Engineering Challenges of Solution and Slurry-Phase Chemical Hydrogen Storage Materials for Automotive Fuel Cell Applications"

_molecules, 2021, doi:10.3390/molecules26061722_

Round 1
Reviewer 1 Report
This manuscript provides useful information for hydrogen storage researchers and should be published. My comments have to do with the presentation itself, which is in need of significant improvement. The authors need to keep in mind that some readers are unfamiliar with certain terms that may be considered to be common knowledge within a quite restricted community. Readers should not be expected to look up terms, and some terms cannot even be easily looked up. The designation of various chemicals is haphazard and must be improved. The definitions of various terms must be given when first mentioned, not pages later. Finally, the labeling of many figures is poor, including small, pale blue lettering in some cases. A list is given below.
Specific comments follow.
Page 2
Table 1
Various acronyms need to be defined below the table. TBD—some info should be provided, even if approximate.
Page 3
Alane and ammonia borane should have some chemical formulae. I’ve been a chemist for about 50 years and have never heard of alane. Wikipedia is not very certain about it, saying it may refer to aluminium hydride or a 1997 song! Finally, on page 7, I see a chemical formula for alane but not for ammonia borane, which finally shows up on page 12. On page 8, both alane and ammonium hydride are used interchangeably. It would be better to define the terms and use them consistently.
What is the meaning of “downselect” ? This is a kind of industrial jargon, I think, and should be either defined or replaced. The same comment applies for “upselect.” Also, please use these terms consistently, not down select, down-select and downselect.
“minimum screening criteria which gave the Center a rough assessment of its capabilities. The quantified minimum screening criteria for each materials group, i.e., Metal Hydrides, Chemical Hydrides and Adsorbent Materials, are listed in Table 1.”
Should be Table 2.
Page 4
Table 2. Is $10,000/kg correct—it seems a bit expensive.
Table 4
Please use either ammonia borane or AB, not both. If the latter, please define below the Table.
Page 8
In some cases, the alpha symbol shows up as a strange coil symbol.
Page 12
“...following references [references]...”
Page 14
Define BmimCl.
Page 15
“Develop and demonstrate...” is an incomplete sentence.
Page 16
What is “slugging?” The term needs to be defined, since not all readers will be familiar with it. It is unfair to expect readers to do a google search.
Page 19
Figure 10 axis labels are too small to read. See below for a list of figures in need of improvement.
“Develop and demonstrate...” is an incomplete sentence.
Page 23
Has “hydrogen selectivity” been defined? On page 34, I see a partial definition. On page 40, still another partial definition is given for “gas-phase reaction selectivity.” Is that the same thing or not? It would be best to give a complete definition when the term is used first and then to refer to the concept with the exact same wording thereafter. Also, “unity” is preferable to “one.”
Page 27
Details are needed for IoLiLyte ionic liquid (and others).
“Investigate potential solvent...” is an incomplete sentence.
Page 28
“Silicon oil” —> “silicone oil”
Page 29
“Determine fuel cell tolerance” is an incomplete sentence.
Page 30
“Develop and demonstrate...” is an incomplete sentence.
Page 37
Here, the objective is given as a full sentence. This style should be used throughout.
Page 38
“Boilerplate” is industrial slang and should be defined or replaced.
Page 41
Table 6. This is very difficult to read without expansion.
Page 42
The authors have already used 42 pages of text. Why is it impossible to add another half page or so of COE collaborator names?
Figures that must be improved, particularly increasing font size: 8, 9, 10, 16 (extreme lack of labeling), 18, 27, 30.
Author Response
We have addressed all comments in the revised manuscript
Reviewer 2 Report
The paper is a very interesting collection (as declared by the authors themselves) of innovative results concerning liquid-phase and slurry-phase chemical hydrogen storage media and their potential as future hydrogen storage systems for automotive applications. The reported results are original and interesting for hydrogen storage people. The data are described in a clear and detailed way. Tables and figures are well edited and clear. The applicative and practical aspects of the work are very well evidenced. I think the paper is suitable for publication.
Author Response
We have addressed all comments from Reviewer 2